# TOWARDS CALIBRATING PROMPT TUNING OF VISION-LANGUAGE MODELS

## ABSTRACT

Prompt tuning of large-scale vision-language models such as CLIP enables efficient task adaptation without updating model weights. However, it often leads to poor confidence calibration and unreliable predictive uncertainty. We address this problem by proposing a calibration framework that enhances predictive reliability while preserving the geometry of the pretrained CLIP embedding space, which is required for robust generalization. Our approach extends the standard cross-entropy loss with two complementary regularizers: (1) a mean–variance margin penalty that stabilizes inter-class logit margins by maximizing their average while minimizing dispersion, mitigating underconfidence and overconfidence spikes; and (2) a text moment-matching loss that aligns the first and second moments of tuned text embeddings with their frozen CLIP counterparts, preserving semantic dispersion crucial for generalization. Through extensive experiments across 7 prompt-tuning methods and 11 diverse datasets, we demonstrate that our approach significantly reduces the Expected Calibration Error (ECE) compared to competitive calibration techniques on both base and novel classes. Our code will be made publicly available.

## 1 INTRODUCTION

Vision language models (VLM), such as CLIP (Radford et al., 2021), have significantly advanced open-vocabulary image recognition by effectively using large-scale natural language supervision. To efficiently adapt these pre-trained models to downstream tasks, parameter–efficient techniques, particularly prompt tuning, have become popular (Liu et al., 2023b). Prompt tuning modifies only a small subset of parameters, substantially enhancing performance on seen (base) classes while preserving the model's inherent generalization ability to unseen (novel) classes (Khattak et al., 2023a; Zhou et al., 2022c). This balance between specialization and generalization has driven the widespread adoption of prompt-tuned VLMs in healthcare, autonomous systems, and industrial applications where recognizing expected and unexpected visual concepts is essential for safe operation (Zhao et al., 2025; Elhenawy et al., 2025).

Despite these advances, existing prompt tuning techniques predominantly prioritize accuracy, often neglecting the critical issue of confidence calibration. Miscalibration occurs when a model's predicted confidence poorly aligns with its actual likelihood of correctness, resulting in unreliable uncertainty estimates (Wang, 2023; Guo et al., 2017). This reliability gap poses substantial challenges for the deployment of prompt-tuned VLMs in applications where incorrect high-confidence predictions can have serious consequences, such as autonomous systems that fail to identify obstacles or medical imaging tools that overlook critical abnormalities (Lambert et al., 2024; Shao et al., 2024). We note that maintaining well-calibrated confidence estimates across both base and novel categories remains largely unexplored, despite being crucial for real-world VLM deployment (Gawlikowski et al., 2023).

Only a few recent efforts have explicitly addressed calibration in the context of prompt-tuned CLIP. DAC (Wang et al., 2024b) implements post-hoc temperature scaling for novel classes based on semantic distances between class embeddings. However, this method cannot constrain how prompt tuning alters the original embedding space, allowing problematic transformations like embedding collapse or clustering that introduce spurious semantic relationships. Consequently, the model makes overconfident predictions for novel inputs that fall near distorted decision boundaries. Similarly, (Murugesan et al., 2024) attempt to normalize output logits to match zero-shot CLIP's distribution

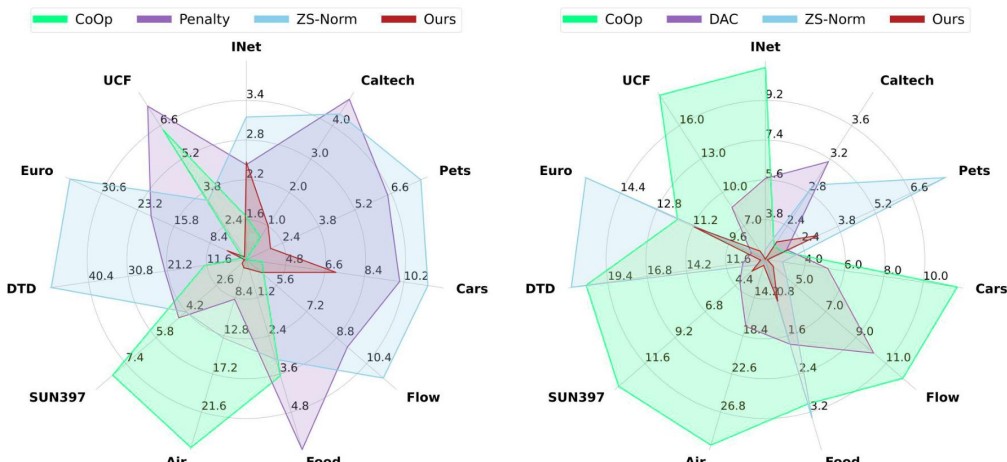

Figure 1: **Expected Calibration Error (ECE) on 11 datasets with CoOp** (Zhou et al., 2022b) shown as radar plots. *Left*: Base classes—our method (red) consistently yields lower ECE than competing approaches, with notable gains on DTD, EuroSat, and Food. *Right*: Novel classes—our method reduces miscalibration relative to vanilla CoOp (yellow) and outperforms DAC (Wang et al., 2024a) and ZS-Norm (Murugesan et al., 2024), especially on Aircraft and Cars. The uniformly smaller footprint of our curve indicates superior calibration, supporting the effectiveness of the proposed dual-regularization approach in addressing both underconfidence on base classes and overconfidence on novel classes.

characteristics. While this approach adjusts the global statistical properties of model outputs, it often fails to preserve the inter-class relationships in the embedding space. This limitation prevents the method from effectively addressing both underconfidence on base classes and overconfidence on novel classes simultaneously. We address the dual calibration problem in prompt-tuned CLIP with a training-time regularization framework that preserves pretrained semantics while stabilizing predictive margins. Our approach has two complementary components that jointly target underconfidence on base classes and overconfidence on novel classes. Specifically, our contributions are:

- We propose a **mean-variance margin regularization** that shapes logit distributions by encouraging sufficiently large margins between correct and incorrect predictions while constraining margin variability to prevent spurious confidence spikes.

- We introduce a **text moment-matching loss** that preserves the geometric structure of CLIP's pretrained embedding space by aligning the statistical moments of tuned and frozen text embeddings. This preserves critical semantic relationships without restricting task-specific adaptations.

- We evaluate our approach across 11 diverse datasets and 7 prompt-tuning frameworks, demonstrating consistent improvements in calibration without compromising accuracy, outperforming post-hoc and training-time baselines (see Figure 1).

*Importantly, our method is agnostic to the underlying prompt tuning technique, does not require additional inference time computation and functions as a plug-and-play module for existing frameworks.*

## 2 RELATED WORK

**Prompt Tuning for Vision-Language Models.** Prompt tuning adapts VLMs like CLIP by learning a small set of text tokens while freezing the image/text encoders, enabling parameter-efficient specialization with minimal supervision. Early work explored static prompts shared across all inputs (Zhou et al., 2022d), while subsequent extensions proposed instance-conditioned prompts that adapt dynamically to each image (Zhou et al., 2022c). Further advances have incorporated multi-modal prompt learning (Khattak et al., 2023a) and visual context modulation. These methods are typically trained with few-shot supervision on a set of *base* classes and evaluated for zero-shot

generalization on *novel* classes, aiming to improve open-vocabulary recognition without fine-tuning the backbone. *However, these existing methods have been focusing on improving classification accuracy, with little attention paid to the calibration of predicted probabilities.*

**Calibration of Deep Neural Networks.** Traditional post-hoc calibration methods, such as Temperature Scaling (Guo et al., 2017), calibrate confidence scores by applying a scalar temperature to the model's logits, typically learned on a held-out validation set. While effective under in-distribution settings, these methods assume access to labeled data from the target domain and often fail to generalize to out-of-distribution scenarios, where such supervision is unavailable (Niculescu-Mizil & Caruana, 2005; Wang et al., 2024b). On the other hand, train-time calibration approaches integrate auxiliary loss terms during model training that penalize miscalibrated predictions (Kumar et al., 2018; Ovadia et al., 2019), yielding models with more reliable uncertainty estimates. *However, these methods typically require fully labeled datasets and involve fine-tuning the entire model, rendering them unsuitable for settings like prompt tuning, which operate in a few-shot regime and update only a small number of parameters.*

**Calibrating Prompt Tuning.** Recent work has begun to address calibration in prompt-tuned VLMs. DAC (Wang et al., 2024b) applies post-hoc temperature scaling based on semantic distances between class embeddings but often degrade the sharp decision boundaries of their pretrained counterparts. Concurrently, (Murugesan et al., 2024) tackled calibration within the prompt tuning framework by identifying expanded logit distributions as a key issue and introducing zero-shot normalization and sample-adaptive logit scaling to restore alignment with zero-shot CLIP. Methods aiming to improve calibration for test-time prompt tuning (Yoon et al., 2024; Sharifdeen et al., 2025) have aimed to enhance the dispersion of text features, but require additional computation inference. These efforts reveal key insights: preserving CLIP's embedding geometry is essential for novel class calibration, sufficient class separation helps prevent overconfidence, and post-hoc corrections cannot fully restore pretrained calibration properties. *Our method unifies these insights by directly constraining embedding transformations during training while simultaneously handling the distinct miscalibration patterns of base and novel classes, without compromising semantic relationships.*

## 3 METHOD

### 3.1 PRELIMINARIES

**Zero-Shot Inference for CLIP.** CLIP enables zero-shot classification by learning a joint embedding space for images and natural language descriptions through large-scale contrastive pretraining. The model comprises an image encoder $\mathbf{E}_{\text{img}} : \mathcal{I} \to \mathbb{R}^d$ and a text encoder $\mathbf{E}_{\text{txt}} : \mathcal{T} \to \mathbb{R}^d$, where $\mathcal{I}$ and $\mathcal{T}$ denote the image and text spaces, respectively. During inference, given an input image $\mathbf{I} \in \mathcal{I}$, the image encoder produces a feature embedding $\mathbf{v} = \mathbf{E}_{\text{img}}(\mathbf{I})$. To perform classification, CLIP compares this visual embedding against textual representations of candidate class labels. Specifically, each class $y_i \in \{y_1, \ldots, y_K\}$ is converted into text prompts using a fixed prompt template (e.g., $\mathbf{t}(y_i) = $ "A photo of a {class}"), and encoded by the text encoder as $\mathbf{u}_i = \mathbf{E}_{\text{txt}}(\mathbf{t}(y_i))$. The similarity between image and text embeddings is computed by cosine similarity as $s_i = \cos(\mathbf{v}, \mathbf{u}_i)$, and the predicted class probabilities are obtained using a temperature-scaled softmax: $\mathbb{P}(y_i \mid \mathbf{I}) = \exp(\tau s_i) / \sum_{j=1}^{K} \exp(\tau s_j)$, where $\tau$ is softmax temperature parameter. The predicted label $\hat{y}$ and its associated confidence $\hat{p}$ are given by $\hat{y} = \arg\max_i \mathbb{P}(y_i \mid \mathbf{I})$ and $\hat{p} = \max_i \mathbb{P}(y_i \mid \mathbf{I})$, respectively.

**Prompt Learning for CLIP.** While zero-shot classification with handcrafted templates is effective, it may not provide optimal task-specific context. Prompt learning addresses this by optimizing the prompt tokens directly for downstream performance. Instead of using fixed text templates, a set of learnable tokens $\mathcal{T} = \{\mathbf{p}_1, \ldots, \mathbf{p}_M\}$ is introduced as a prefix to the class name in each prompt. For any class $y \in \mathcal{Y}$, the composed prompt is defined as $\mathbf{t}(y) = [\mathbf{p}_1, \ldots, \mathbf{p}_M, \mathbf{e}_y]$, where $\mathbf{e}_y$ is a static embedding of the class name. The text encoder maps this into a class representation $\mathbf{c}_y = \mathbf{E}_{\text{txt}}(\mathbf{t}(y))$. Given an image $\mathbf{x}$, the logit for class $y$ is computed as $s_y = \tau \cdot \cos(\mathbf{E}_{\text{img}}(\mathbf{x}), \mathbf{c}_y)$, and class probabilities are obtained via softmax as in the zero-shot case. The prompt tokens $\mathcal{T}$ are optimized using a small labeled training set $\mathcal{D} = \{(\mathbf{x}_i, y_i)\}_{i=1}^{N}$, typically by minimizing the cross-entropy loss over predicted class probabilities. We denote the set of classes seen during prompt tuning as $\mathcal{Y}_{\text{base}} \subset \mathcal{Y}$, and the remaining classes encountered at test time as $\mathcal{Y}_{\text{novel}} = \mathcal{Y} \setminus \mathcal{Y}_{\text{base}}$.

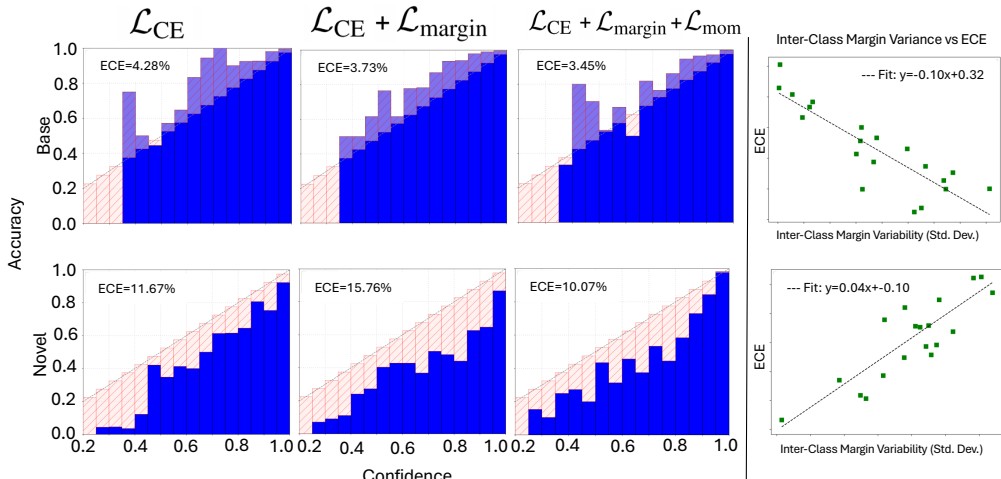

Figure 2: **Dual miscalibration in prompt-tuned CLIP.** *Left (top row)*: Base classes are underconfident (accuracy exceeds confidence) that improves with our regularization terms. *Left (bottom row)*: Novel classes exhibit overconfidence (confidence exceeds accuracy) that our method effectively mitigates. (*Right*) Inter-class margin variability vs. ECE shows a *negative* correlation for base classes and a *positive* correlation for novel classes, indicating that prompt tuning tightens margins on base classes and inflates them on novel classes, degrading reliability. These trends motivate our margin-stabilizing and moment-preserving regularizers.

### 3.2 PROPOSED METHOD

Our goal is to improve the reliability of prompt-tuned CLIP by ensuring well-calibrated confidence estimates across *both base and novel classes*. This dual calibration challenge arises because prompt tuning introduces asymmetric boundary distortions: reduced logit margins for base classes (causing underconfidence) and inflated margins for novel classes (causing overconfidence). Unlike existing methods (Murugesan et al., 2024; Wang et al., 2024b) that address either base class underconfidence or novel class overconfidence in isolation, our approach simultaneously tackles both through: (1) margin-based regularization that encourages more discriminative decision boundaries, and (2) moment-matching loss that preserves CLIP's well-calibrated embedding structure. Next, we first analyze miscalibration in prompt-tuned CLIP and then present a training-time solution based on margin regularization and moment matching.

**Dual Miscalibration in Fine-tuned CLIP.** To investigate calibration issues in prompt-tuned CLIP, we systematically analyzed calibration behavior across diverse prompt configurations and datasets. Figure 2 demonstrates that dual calibration problem through reliability diagrams and margin analysis. For **base classes**, reliability diagrams show underconfidence: predicted probabilities trail behind actual accuracy, reflecting reduced margins between the top-1 and runner-up classes. Larger margins correspond to lower calibration error, confirming the link between boundary tightness and under-confidence. For **novel classes**, the opposite pattern emerges: predictions are overconfident, with inflated margins driving calibration error upward. The scatter plots in Figure 2 (right) quantify these complementary trends, revealing a negative correlation between margin variability and calibration error for base classes, and a positive correlation for novel classes.

These findings indicate that prompt tuning disrupts the naturally well-calibrated decision boundaries of zero-shot CLIP. An ideal calibrated open-vocabulary model should maintain class-appropriate confidence margins across both base and novel categories while preserving the semantic geometry of the pretrained text embedding space. This motivates our framework that explicitly stabilizes decision margins and preserves pretrained semantics.

#### 3.2.1 MEAN-VARIANCE MARGIN REGULARIZATION

To address the dual miscalibration in prompt-tuned CLIP, we introduce a mean-variance margin regularization that shapes logit distributions during training. Our mean-variance loss maintains sufficiently large margins between correct and incorrect predictions to prevent base class underconfidence while enforcing margin consistency across samples to avoid novel class overconfidence. Formally, for a batch of samples $\{(\mathbf{x}_i, y_i)\}_{i=1}^{B}$, let $\mathbf{z}_i$ denote the predicted logits for sample $i$. We define the

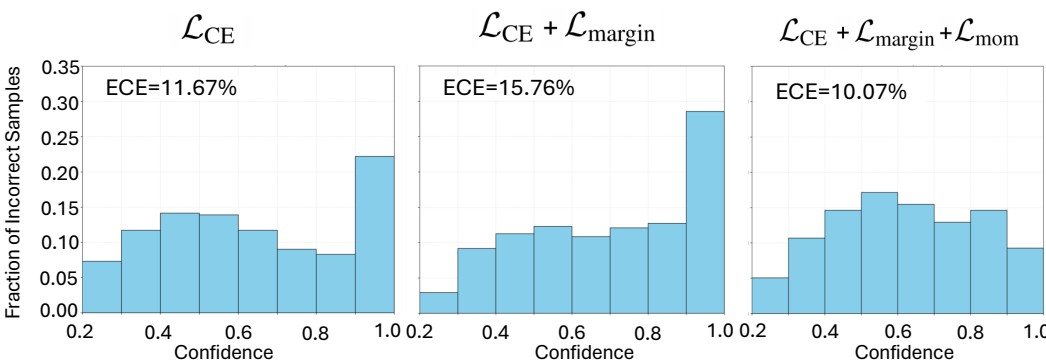

Figure 3: **Errors by confidence on novel classes.** Higher error mass in high-confidence bins indicates overconfidence. Both **Cross-Entropy** and **Cross-Entropy + Margin** place more misclassified samples in high-confidence regions, whereas adding **Text Moment-Matching** to the **Margin** term shifts errors away from these bins, reducing overconfidence.

per-sample logit margin as the difference between the ground-truth class logit and the highest logit among incorrect classes as $m_i = z_{i,y_i} - \max_{j \neq y_i} z_{i,j}$. The *mean-variance margin regularization* loss is then defined as:

$$\mathcal{L}_{\text{Margin}} = -\alpha \cdot \frac{1}{B} \sum_{i=1}^{B} m_i + \beta \cdot \text{Var}(m_1, \ldots, m_B),$$

where $\alpha, \beta > 0$ are hyperparameters that control the trade-off between maximizing the average margin and minimizing its variance across the batch.

The dual design creates complementary objectives that act synergistically: the mean term (weighted by $\alpha$) promotes sufficient separation for confident base class predictions, while the variance term (weighted by $\beta$) prevents margin inconsistency that creates overconfident novel class predictions. Without variance regularization, models may develop erratic decision boundaries with spurious confidence spikes on novel classes. Without mean regularization, uniformly small margins cause systematic underconfidence on base classes, as shown in Figure 5. Unlike prior margin-based approaches such as MBLS (Liu et al., 2023a) that impose hard per-sample constraints, our batch-level statistical approach avoids over-regularization while maintaining adaptability across diverse class distributions, yielding more reliable confidence estimates in open-vocabulary settings.

### 3.2.2 TEXT MOMENT-MATCHING LOSS

While the margin regularizer shapes confidence near decision boundaries, it operates on logits and does not directly constrain the geometry of the text embedding space. Empirically (Figure. 2), using the margin term alone can *increase* ECE on novel classes: when the top-1 prediction is incorrect, maximizing the margin widens the gap to the runner-up for the *wrong* class, amplifying overconfidence. To maintain calibrated generalization on novel classes, we therefore propose to preserve the global semantic relationships between class text embeddings. To this end, we introduce a *text moment-matching* objective that aligns the statistical properties of the prompt-tuned text embeddings with those of frozen CLIP. Let $\{\tilde{\mathbf{c}}_y\}_{y \in \mathcal{B}}$ denote the prompt-tuned text embeddings for a batch of classes $\mathcal{B} \subset \mathcal{Y}_{\text{base}}$, and $\{\mathbf{c}_y^0\}_{y \in \mathcal{B}}$ the corresponding frozen (zero-shot) embeddings. We match both the first- and second-order moments of these sets:

$$\mu_{\tilde{c}} = \frac{1}{|\mathcal{B}|} \sum_{y \in \mathcal{B}} \tilde{\mathbf{c}}_y, \qquad\qquad \mu_{c^0} = \frac{1}{|\mathcal{B}|} \sum_{y \in \mathcal{B}} \mathbf{c}_y^0, \tag{1}$$

$$\Sigma_{\tilde{c}} = \frac{1}{|\mathcal{B}|} \sum_{y \in \mathcal{B}} (\tilde{\mathbf{c}}_y - \mu_{\tilde{c}})(\tilde{\mathbf{c}}_y - \mu_{\tilde{c}})^\top, \qquad \Sigma_{c^0} = \frac{1}{|\mathcal{B}|} \sum_{y \in \mathcal{B}} (\mathbf{c}_y^0 - \mu_{c^0})(\mathbf{c}_y^0 - \mu_{c^0})^\top. \tag{2}$$

The moment-matching loss constrains both distributional center and spread:

$$\mathcal{L}_{\text{mom}} = \|\mu_{\tilde{c}} - \mu_{c^0}\|_2^2 + \|\Sigma_{\tilde{c}} - \Sigma_{c^0}\|_F^2.$$

Minimizing $\mathcal{L}_{\text{mom}}$ preserves the semantic center and dispersion of the frozen CLIP space, which supports generalization and curbs high-confidence errors introduced by prompt-induced drift. Unlike direct $\ell_1/\ell_2$ alignment, which forces rigid instance-level correspondence and can hinder adaptation, moment matching constrains only global distributional statistics, keeping local task-specific prompt adjustments expressive.

**Complementarity with the margin loss.** The margin regularizer (logit space) increases inter-class separation but, when the top-1 class is incorrect, can enlarge the wrong gap and worsen novel-class overconfidence. Moment loss (embedding space) counterbalances this by preserving CLIP's semantic geometry, maintaining relative class structure and angular spread, thus curbing such failure modes. Together, the two terms act synergistically: the margin term enforces discriminability; the moment term stabilizes geometry, yielding calibrated decision boundaries for both base and novel classes without sacrificing downstream performance (see Figure 3).

Finally, we combine the two regularizers with cross-entropy to form the full objective:

$$\mathcal{L}_{\text{total}} = \mathcal{L}_{\text{CE}} + \lambda_{\text{Margin}} \, \mathcal{L}_{\text{Margin}} + \lambda_{\text{mom}} \, \mathcal{L}_{\text{mom}}, \tag{3}$$

where $\lambda_{\text{Margin}}, \lambda_{\text{mom}} \geq 0$ controls the strength of each term. This joint optimization addresses both aspects of the dual calibration problem: reducing underconfidence in base classes and overconfidence in novel classes while preserving task-specific adaptation. The terms are complementary: the margin loss enforces discriminability and the moment loss preserves semantic geometry, allowing prompt-tuned models to inherit the well-calibrated behavior of zero-shot CLIP.

## 4 EXPERIMENTS

We follow the standard few-shot protocol (Zhou et al., 2022b), each dataset is split into disjoint base and novel classes. Prompt tuning is performed only on base classes using a limited number of labeled samples per class. The calibration performance is reported for both base and novel classes.

**Datasets.** We evaluate on 11 datasets spanning coarse-grained, fine-grained, and domain-specific tasks. Object classification is assessed on ImageNet (Deng et al., 2009) and Caltech101 (Fei-Fei et al., 2004). Fine-grained recognition tasks include DTD (Cimpoi et al., 2014) (textures), Flowers (FLW) (Nilsback & Zisserman, 2008), Food101 (Bossard et al., 2014), SUN397 (Xiao et al., 2016), and UCF101 (Soomro et al., 2012). Domain-specific benchmarks comprise Stanford Cars (Krause et al., 2013), FGVC-Aircraft (Maji et al., 2013), Oxford Pets (Parkhi et al., 2012), and EuroSAT (Helber et al., 2019). We also provide results on out-of-distribution datasets for the ImageNet-A (Hendrycks et al., 2021b), ImageNet-V2 (Shankar et al., 2020), ImageNet-R (Hendrycks et al., 2021a), and ImageNet-S (Wang et al., 2019) datasets in the suppl. material.

**Baseline Methods.** We benchmark our approach against a diverse and comprehensive set of prompt-tuning and calibration techniques. Classical baselines include MBLS (Liu et al., 2023a) and Temperature Scaling (Guo et al., 2017). We also compare with recent methods such as DAC (Wang et al., 2024a), a post-hoc approach for adapting CLIP to novel classes, and a training-time regularization method introduced by (Murugesan et al., 2024). For prompt tuning, we evaluate three representative approaches in the main paper: CoOp (Zhou et al., 2022b), KgCoOp (Yao et al., 2023a), and MaPLe (Khattak et al., 2023b). Results for additional methods including CoCoOp (Zhou et al., 2022a), ProDA (Yao et al., 2023b), ProGrad (Zhu et al., 2023), and PromptSRC (Khattak et al., 2023c) are provided in the supp. material.

**Evaluation Metrics and Implementation details.** We evaluate classification performance using top-1 accuracy (ACC). To measure model calibration, we report the Expected Calibration Error (ECE), a widely used metric. We also report results with Adaptive Calibration Error (Nixon et al., 2019) (ACE) and Maximum Calibration Error (Naeini et al., 2015) (MCE) in the supp. material. For all experiments, we use CLIP (ViT-B/16) (Radford et al., 2021) as the pre-trained vision-language model. Prompt-tuning is conducted in a few-shot setting with 16 samples per class, using a learning rate of 0.005 and a batch size of 8. For each baseline method, we adopt its official implementation. All experiments are performed on an NVIDIA RTX A6000 GPU with 48GB memory. We provide detailed hyperparameters in the supp. material.

**Calibration on Base Classes.** Table 1 reports the results on 11 fine-grained classification datasets using three prompt-tuning methods. Our approach consistently maintains or slightly improves the

Table 1: **Accuracy and calibration performance on base classes across 11 fine-grained classification benchmarks**. Top-1 accuracy (Acc) and Expected Calibration Error (ECE) for multiple prompt-tuning strategies and diverse calibration baselines. Higher Acc. indicates better classification performance, while lower ECE reflects better calibration.

| Method | | INet | Calt | Pets | Cars | Flow | Food | Air | SUN | DTD | Euro | UCF | Avg |
|---|---|---|---|---|---|---|---|---|---|---|---|---|---|
| Zero Shot | Acc. | 72.40 | 97.20 | 91.30 | 63.60 | 71.80 | 90.10 | 27.70 | 69.40 | 53.00 | 57.00 | 71.00 | 69.50 |
| | ECE | 1.51 | 6.49 | 2.25 | 3.74 | 3.11 | 1.57 | 3.03 | 1.59 | 4.53 | 8.35 | 3.24 | 3.58 |
| **CoOp** (Zhou et al., 2022b) | | | | | | | | | | | | | |
| CoOp (Zhou et al., 2022b) | Acc. | 75.60 | 97.98 | 94.77 | 76.22 | 90.00 | 90.20 | 35.23 | 81.14 | 76.27 | 90.24 | 83.32 | 81.00 |
| | ECE | 1.65 | 0.66 | 1.00 | 3.73 | 4.93 | 3.66 | 25.70 | 8.11 | 12.17 | 1.75 | 6.44 | 6.35 |
| MBLS (Liu et al., 2023a) | Acc. | 75.12 | 97.89 | 91.11 | 76.21 | 89.34 | 89.78 | 34.32 | 81.32 | 76.34 | 90.12 | 82.78 | 80.39 |
| | ECE | 2.98 | 9.6 | 7.70 | 12.20 | 5.69 | 12.34 | 10.48 | 16.80 | 4.25 | 8.02 | 9.39 | 9.04 |
| Temp. Scal. (Guo et al., 2017) | Acc. | 75.60 | 98.19 | 94.15 | 78.65 | 97.72 | 90.10 | 42.00 | 81.32 | 80.67 | 90.70 | 84.56 | 83.06 |
| | ECE | 1.50 | 1.20 | 2.54 | 6.63 | 4.60 | 0.50 | 3.43 | 2.01 | 3.86 | 4.76 | 1.57 | 2.96 |
| DAC (Wang et al., 2024a) | Acc. | - | - | - | - | - | - | - | - | - | - | - | - |
| | ECE | - | - | - | - | - | - | - | - | - | - | - | - |
| ZS-Norm (Murugesan et al., 2024) | Acc. | 76.10 | 97.85 | 94.38 | 77.78 | 95.76 | 89.52 | 39.74 | 81.37 | 81.02 | 90.45 | 84.01 | 82.54 |
| | ECE | 3.15 | 4.35 | 7.75 | 11.30 | 11.29 | 3.14 | 13.05 | 4.22 | 49.53 | 37.04 | 3.47 | 13.48 |
| Penalty (Murugesan et al., 2024) | Acc. | 76.44 | 97.72 | 95.11 | 77.05 | 96.30 | 87.92 | 38.07 | 81.04 | 77.32 | 47.09 | 80.47 | 77.68 |
| | ECE | 2.43 | 4.79 | 6.47 | 10.01 | 9.38 | 5.98 | 8.59 | 4.59 | 21.84 | 20.47 | 7.42 | 9.27 |
| **Ours** | Acc. | 76.53 | 98.06 | 94.95 | 77.32 | 97.21 | 90.38 | 38.62 | 81.68 | 80.44 | 88.56 | 84.68 | 82.58 |
| | ECE | 2.47 | 1.01 | 1.94 | 7.10 | 4.80 | 0.30 | 4.96 | 1.22 | 2.42 | 4.90 | 1.11 | 2.93 |
| **MaPLe** (Khattak et al., 2023b) | | | | | | | | | | | | | |
| MaPLe (Khattak et al., 2023b) | Acc. | 76.71 | 97.97 | 95.53 | 72.93 | 96.00 | 90.80 | 36.33 | 80.55 | 79.63 | 91.13 | 83.20 | 82.41 |
| | ECE | 2.27 | 1.54 | 2.68 | 7.25 | 4.28 | 1.27 | 3.86 | 1.27 | 4.18 | 3.42 | 2.68 | 3.19 |
| MBLS (Liu et al., 2023a) | Acc. | 75.59 | 98.23 | 95.23 | 72.77 | 95.93 | 90.80 | 36.20 | 80.73 | 80.03 | 90.93 | 84.13 | 82.50 |
| | ECE | 29.06 | 5.03 | 6.64 | 19.06 | 12.74 | 6.55 | 5.60 | 11.01 | 4.79 | 3.73 | 8.46 | 8.36 |
| Temp. Scal. (Guo et al., 2017) | Acc. | 76.66 | 97.97 | 94.93 | 72.70 | 95.93 | 90.63 | 36.37 | 80.73 | 78.60 | 93.60 | 84.00 | 82.55 |
| | ECE | 2.37 | 1.26 | 2.28 | 4.96 | 3.44 | 0.71 | 3.04 | 2.84 | 5.98 | 1.31 | 3.07 | 2.89 |
| DAC (Wang et al., 2024a) | Acc. | - | - | - | - | - | - | - | - | - | - | - | - |
| | ECE | - | - | - | - | - | - | - | - | - | - | - | - |
| ZS-Norm (Murugesan et al., 2024) | Acc. | 76.63 | 97.57 | 95.70 | 73.07 | 95.63 | 90.57 | 36.00 | 80.97 | 80.43 | 91.30 | 83.87 | 82.51 |
| | ECE | 1.64 | 23.30 | 5.91 | 8.66 | 11.49 | 1.13 | 7.87 | 2.33 | 7.02 | 19.38 | 3.86 | 9.10 |
| Penalty (Murugesan et al., 2024) | Acc. | 76.72 | 98.07 | 95.30 | 72.43 | 95.77 | 90.73 | 34.33 | 80.93 | 64.60 | 36.77 | 83.03 | 75.20 |
| | ECE | 3.87 | 5.41 | 6.37 | 13.53 | 12.67 | 3.87 | 8.42 | 7.28 | 19.97 | 13.43 | 8.50 | 9.95 |
| **Ours** | Acc. | 76.72 | 97.97 | 94.93 | 72.80 | 96.20 | 90.43 | 36.80 | 81.10 | 80.73 | 92.00 | 84.50 | 82.75 |
| | ECE | 2.39 | 1.19 | 1.54 | 7.92 | 3.45 | 0.65 | 4.50 | 1.55 | 3.56 | 1.33 | 2.12 | 2.78 |
| **KGCoOp** (Yao et al., 2023a) | | | | | | | | | | | | | |
| KGCoOp (Yao et al., 2023a) | Acc. | 75.75 | 97.70 | 94.68 | 72.70 | 95.16 | 90.57 | 36.77 | 80.59 | 79.40 | 86.14 | 83.51 | 81.18 |
| | ECE | 2.52 | 2.92 | 3.27 | 10.16 | 12.12 | 1.68 | 3.27 | 4.92 | 8.39 | 11.90 | 5.03 | 6.02 |
| MBLS (Liu et al., 2023a) | Acc. | 76.23 | 97.81 | 95.00 | 75.34 | 96.24 | 90.49 | 38.28 | 80.86 | 79.94 | 87.96 | 83.45 | 81.96 |
| | ECE | 6.19 | 4.30 | 5.26 | 13.43 | 12.48 | 4.08 | 8.01 | 8.16 | 9.03 | 11.97 | 5.86 | 8.07 |
| Temp. Scal. (Guo et al., 2017) | Acc. | 75.77 | 97.66 | 94.67 | 70.08 | 94.65 | 90.50 | 35.81 | 80.51 | 78.74 | 86.44 | 83.32 | 80.74 |
| | ECE | 6.47 | 4.16 | 5.13 | 11.70 | 15.35 | 3.64 | 7.41 | 8.50 | 11.12 | 15.79 | 7.39 | 8.79 |
| DAC (Wang et al., 2024a) | Acc. | - | - | - | - | - | - | - | - | - | - | - | - |
| | ECE | - | - | - | - | - | - | - | - | - | - | - | - |
| ZS-Norm (Murugesan et al., 2024) | Acc. | 75.78 | 94.14 | 97.65 | 74.55 | 73.90 | 91.71 | 30.79 | 76.50 | 51.49 | 65.39 | 76.44 | 73.49 |
| | ECE | 2.70 | 1.65 | 3.51 | 3.85 | 4.72 | 2.20 | 8.42 | 3.23 | 6.37 | 6.16 | 3.83 | 4.24 |
| Penalty (Murugesan et al., 2024) | Acc. | 75.65 | 97.70 | 94.68 | 72.45 | 93.86 | 90.59 | 37.76 | 80.63 | 78.40 | 83.09 | 82.97 | 80.71 |
| | ECE | 2.73 | 3.27 | 3.22 | 10.58 | 13.01 | 1.73 | 9.59 | 6.51 | 20.40 | 6.51 | 6.07 | 7.57 |
| **Ours** | Acc. | 75.84 | 97.68 | 94.84 | 71.65 | 95.22 | 90.52 | 36.03 | 80.70 | 78.47 | 85.10 | 83.16 | 80.34 |
| | ECE | 2.14 | 1.88 | 2.96 | 8.10 | 11.21 | 1.12 | 4.81 | 4.12 | 7.01 | 12.64 | 4.14 | 5.47 |

classification accuracy while significantly reducing the calibration error. When applied to MaPLe, it improves the average accuracy from 82.41% to 82.75% and reduces ECE from 3.19% to 2.78%. For CoOp, we observe a larger drop in ECE from 6. 35% to 2. 93%, surpassing both post hoc techniques like temperature scaling (2.96%) and prior regularization methods. Gains are especially pronounced on highly miscalibrated datasets; for example, CoOp on Aircraft shows a reduction from 25.70% to 4.96% in ECE. These improvements demonstrate that preserving margin distributions and embedding geometry effectively mitigates underconfidence in base-class predictions.

**Calibration on Novel Classes.** Table 2 reports accuracy and ECE on novel classes under the open-vocabulary setting. While prompt-tuned models such as CoOp and MaPLe offer improved accuracy over zero-shot CLIP, they often exhibit severe overconfidence (e.g., CoOp ECE = 12.45, MaPLe ECE = 5.76). Post-hoc methods like Temperature Scaling and DAC yield marginal improvements but struggle to correct the underlying semantic drift. Our method consistently reduces calibration error while preserving or improving accuracy. For example, with MaPLe, our method reduces average ECE from 5.76 to 4.23 while maintaining comparable accuracy (75.14%). Notably, our method consistently outperforms recent post-hoc techniques like DAC and avoids the accuracy-calibration tradeoffs seen with methods like ZS-Norm. These results confirm that our approach effectively mitigates overconfidence on novel classes while preserving generalization capabilities.

Table 2: **Accuracy and calibration performance on novel classes across 11 fine-grained classification benchmarks**. We report top-1 accuracy (Acc) and Expected Calibration Error (ECE) for multiple prompt-tuning strategies and diverse calibration baselines.

| Method | | INet | Calt | Pets | Cars | Flow | Food | Air | SUN | DTD | Euro | UCF | Avg |
|---|---|---|---|---|---|---|---|---|---|---|---|---|---|
| Zero Shot | Acc. | 72.40 | 94.10 | 97.10 | 75.00 | 77.50 | 91.10 | 35.90 | 75.50 | 60.60 | 63.80 | 78.60 | 74.30 |
| | ECE | 2.09 | 1.55 | 3.42 | 3.31 | 4.91 | 1.83 | 6.55 | 3.48 | 6.86 | 9.12 | 5.52 | 4.43 |
| **CoOp** (Zhou et al., 2022b) | | | | | | | | | | | | | |
| CoOp (Zhou et al., 2022b) | Acc. | 59.07 | 94.18 | 96.49 | 65.29 | 69.90 | 90.57 | 24.79 | 70.77 | 52.98 | 64.68 | 62.83 | 68.32 |
| | ECE | 10.69 | 2.16 | 1.67 | 11.73 | 12.13 | 3.03 | 30.44 | 13.70 | 20.82 | 11.88 | 18.74 | 12.45 |
| MBLS (Liu et al., 2023a) | Acc. | 59.11 | 95.1 | 96.23 | 65.28 | 69.89 | 90.23 | 24.80 | 70.12 | 53.12 | 64.65 | 62.97 | 68.31 |
| | ECE | 4.09 | 2.21 | 3.45 | 9.7 | 18.9 | 13.8 | 10.2 | 9.7 | 8.9 | 12.1 | 13.21 | 9.66 |
| Temp. Scaling (Guo et al., 2017) | Acc. | 59.07 | 93.45 | 96.03 | 66.70 | 65.86 | 96.60 | 27.37 | 70.67 | 48.19 | 54.70 | 57.51 | 66.92 |
| | ECE | 7.33 | 3.17 | 3.65 | 5.01 | 8.06 | 1.06 | 18.80 | 6.93 | 20.21 | 15.13 | 14.55 | 9.45 |
| DAC (Wang et al., 2024a) | Acc. | - | - | - | - | - | - | - | - | - | - | - | - |
| | ECE | 5.67 | 3.17 | 1.82 | 5.16 | 10.19 | 1.78 | 17.38 | 4.05 | 10.48 | 8.62 | 8.67 | 7.00 |
| ZS-Norm (Murugesan et al., 2024) | Acc. | 66.26 | 93.30 | 93.98 | 66.62 | 67.21 | 88.91 | 25.76 | 70.51 | 44.08 | 50.42 | 62.52 | 66.32 |
| | ECE | 2.46 | 2.89 | 7.94 | 2.87 | 4.41 | 3.32 | 10.18 | 2.47 | 21.80 | 15.93 | 4.28 | 7.14 |
| Penalty (Murugesan et al., 2024) | Acc. | 66.71 | 92.87 | 96.14 | 68.11 | 68.65 | 78.34 | 29.29 | 71.65 | 40.78 | 41.44 | 67.53 | 65.59 |
| | ECE | 2.36 | 2.52 | 7.42 | 2.73 | 4.93 | 4.70 | 7.81 | 2.79 | 4.20 | 13.11 | 4.66 | 5.20 |
| Ours | Acc. | 67.03 | 93.56 | 97.36 | 69.49 | 71.63 | 90.84 | 30.83 | 70.03 | 48.07 | 56.70 | 66.49 | 69.28 |
| | ECE | 2.02 | 2.21 | 3.03 | 2.10 | 3.51 | 0.87 | 10.64 | 3.08 | 9.31 | 11.15 | 4.75 | 4.79 |
| **MaPLe** (Khattak et al., 2023b) | | | | | | | | | | | | | |
| MaPLe (Khattak et al., 2023b) | Acc. | 70.50 | 95.10 | 97.85 | 73.57 | 72.80 | 92.10 | 34.53 | 78.20 | 58.47 | 75.90 | 77.85 | 75.17 |
| | ECE | 1.93 | 1.62 | 2.63 | 3.09 | 11.67 | 1.19 | 11.24 | 2.21 | 12.16 | 11.68 | 3.98 | 5.76 |
| MBLS (Liu et al., 2023a) | Acc. | 68.47 | 94.17 | 96.97 | 71.93 | 68.93 | 91.40 | 33.77 | 78.10 | 54.70 | 75.97 | 78.23 | 73.88 |
| | ECE | 22.82 | 4.06 | 7.41 | 11.41 | 4.84 | 7.06 | 6.06 | 10.41 | 10.31 | 11.25 | 6.63 | 9.30 |
| Temp. Scaling (Guo et al., 2017) | Acc. | 70.46 | 94.83 | 97.30 | 73.47 | 72.77 | 91.77 | 34.07 | 78.13 | 57.97 | 73.77 | 75.33 | 74.53 |
| | ECE | 1.95 | 2.56 | 2.13 | 4.08 | 12.76 | 0.72 | 19.11 | 5.09 | 16.47 | 8.05 | 7.13 | 7.28 |
| DAC (Wang et al., 2024a) | Acc. | - | - | - | - | - | - | - | - | - | - | - | - |
| | ECE | 2.11 | 1.26 | 2.51 | 2.75 | 11.28 | 1.50 | 9.06 | 1.22 | 8.16 | 8.55 | 2.30 | 4.61 |
| ZS-Norm (Murugesan et al., 2024) | Acc. | 70.63 | 90.30 | 97.23 | 73.30 | 70.03 | 91.83 | 34.07 | 78.47 | 60.70 | 68.13 | 77.80 | 73.86 |
| | ECE | 3.67 | 23.02 | 5.00 | 3.26 | 6.05 | 1.62 | 7.82 | 2.65 | 5.23 | 14.53 | 3.33 | 6.93 |
| Penalty (Murugesan et al., 2024) | Acc. | 70.66 | 93.60 | 97.33 | 73.90 | 70.87 | 91.90 | 34.70 | 78.67 | 45.47 | 36.77 | 76.83 | 70.06 |
| | ECE | 1.49 | 3.25 | 6.23 | 5.94 | 5.76 | 4.27 | 4.92 | 5.70 | 8.47 | 13.35 | 6.07 | 5.95 |
| Ours | Acc. | 70.28 | 94.87 | 97.57 | 75.27 | 73.87 | 91.77 | 36.03 | 78.13 | 61.27 | 67.60 | 79.87 | 75.14 |
| | ECE | 1.74 | 1.42 | 2.29 | 2.60 | 10.07 | 0.86 | 8.33 | 1.11 | 7.37 | 7.45 | 3.32 | 4.23 |
| **KGCoOp** (Yao et al., 2023a) | | | | | | | | | | | | | |
| KGCoOp (Yao et al., 2023a) | Acc. | 69.70 | 94.43 | 97.67 | 74.25 | 75.10 | 91.65 | 36.77 | 76.33 | 54.23 | 64.68 | 75.59 | 73.67 |
| | ECE | 1.84 | 1.71 | 3.42 | 3.36 | 5.03 | 2.04 | 6.06 | 1.66 | 4.38 | 8.67 | 2.65 | 3.71 |
| MBLS (Liu et al., 2023a) | Acc. | 69.14 | 94.32 | 94.24 | 73.01 | 73.90 | 90.49 | 28.87 | 75.75 | 56.28 | 64.27 | 73.84 | 72.19 |
| | ECE | 4.60 | 1.62 | 3.16 | 3.95 | 4.00 | 4.00 | 11.39 | 5.56 | 3.23 | 5.30 | 4.10 | 4.63 |
| Temp. Scaling (Guo et al., 2017) | Acc. | 69.79 | 94.54 | 97.56 | 74.94 | 75.37 | 91.66 | 32.35 | 76.79 | 53.83 | 62.17 | 76.91 | 73.27 |
| | ECE | 5.81 | 1.89 | 4.91 | 6.35 | 4.63 | 4.02 | 5.40 | 6.18 | 3.83 | 7.60 | 6.43 | 5.18 |
| DAC (Wang et al., 2024a) | Acc. | - | - | - | - | - | - | - | - | - | - | - | - |
| | ECE | 4.32 | 1.84 | 3.11 | 3.12 | 5.90 | 1.94 | 11.78 | 1.67 | 7.09 | 6.59 | 2.69 | 4.47 |
| ZS-Norm (Murugesan et al., 2024) | Acc. | 69.68 | 94.14 | 97.65 | 74.55 | 73.90 | 91.71 | 30.79 | 76.50 | 51.49 | 65.39 | 76.44 | 72.19 |
| | ECE | 1.80 | 1.65 | 3.51 | 3.85 | 4.72 | 2.20 | 8.42 | 3.23 | 6.37 | 6.16 | 3.83 | 4.16 |
| Penalty (Murugesan et al., 2024) | Acc. | 69.58 | 94.34 | 96.35 | 74.75 | 73.21 | 91.31 | 30.58 | 76.69 | 51.19 | 65.43 | 76.52 | 72.99 |
| | ECE | 1.82 | 1.71 | 4.21 | 3.05 | 5.12 | 2.99 | 8.12 | 4.13 | 5.87 | 6.56 | 3.93 | 4.75 |
| Ours | Acc. | 69.50 | 94.21 | 97.72 | 74.39 | 73.80 | 91.64 | 31.63 | 76.30 | 55.92 | 65.76 | 76.62 | 73.41 |
| | ECE | 1.84 | 1.22 | 3.50 | 3.60 | 4.78 | 1.61 | 7.67 | 1.91 | 3.37 | 4.15 | 3.01 | 3.33 |

| Method | | Calt | Food | DTD | UCF | Flow. | Pets | Air. | Cars | Sun | Euro | Avg |
|---|---|---|---|---|---|---|---|---|---|---|---|---|
| **MaPLe** (Khattak et al., 2023b) | | | | | | | | | | | | |
| $\mathcal{L}_{\text{Margin}}$ | Acc. | 94.80 | 91.87 | 60.47 | 77.30 | 71.70 | 97.83 | 33.93 | 74.80 | 78.33 | 76.40 | 75.74 |
| | ECE | 2.00 | 1.07 | 13.92 | 3.82 | 15.76 | 1.73 | 11.53 | 2.45 | 3.25 | 6.01 | 5.09 |
| $\mathcal{L}_{\text{Margin}} + \mathcal{L}_1$ | Acc. | 94.23 | 92.13 | 59.50 | 79.10 | 73.33 | 97.77 | 26.43 | 74.57 | 78.47 | 70.93 | 74.65 |
| | ECE | 2.32 | 0.55 | 12.13 | 3.90 | 11.71 | 1.72 | 7.39 | 2.58 | 1.58 | 7.80 | 5.17 |
| $\mathcal{L}_{\text{Margin}} + \mathcal{L}_{\text{mom.}}$ | Acc. | 94.87 | 91.77 | 61.27 | 79.87 | 73.87 | 97.57 | 36.03 | 75.27 | 78.13 | 67.60 | 75.63 |
| | ECE | 1.42 | 0.86 | 7.37 | 3.32 | 10.07 | 2.29 | 8.33 | 2.60 | 1.11 | 7.45 | 4.48 |

Table 3: Ablation study on the effect of margin regularization ($\mathcal{L}_{\text{Margin}}$) and moment-matching loss ($\mathcal{L}_{\text{mom.}}$) on novel classes.

**Ablation Study on Loss Components.** Table 3 presents an ablation study analyzing the contribution of each component of our approach on novel class calibration with the MaPLe backbone. All variants build on a cross-entropy baseline, where fine-tuning with only CE yields 5.76% ECE. Adding margin regularization ($\mathcal{L}_{\text{Margin}}$) reduces ECE to 5.09% while maintaining accuracy at 75.74%. When combining margin regularization with direct $\ell_1$ alignment ($\mathcal{L}_1$), we observe improvements on some datasets but slightly lower overall accuracy (74.65%). Our full method, combining margin regularization with moment matching ($\mathcal{L}_{\text{mom}}$), achieves the best calibration (4.48% ECE) while preserving clean accuracy (75.63%). This demonstrates that moment matching provides superior regularization compared to direct embedding alignment, effectively preserving the global structure of CLIP's embedding space while allowing task-specific adaptations.

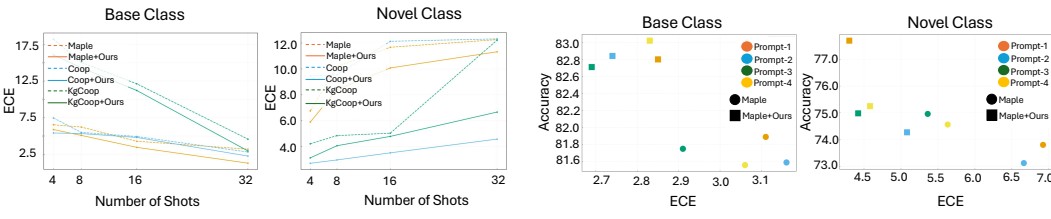

(a) Calibration analyses on number of shots.      (b) Different prompt initialization performance.

Figure 4: Performance analyses on different numbers of shots and hard prompt styles

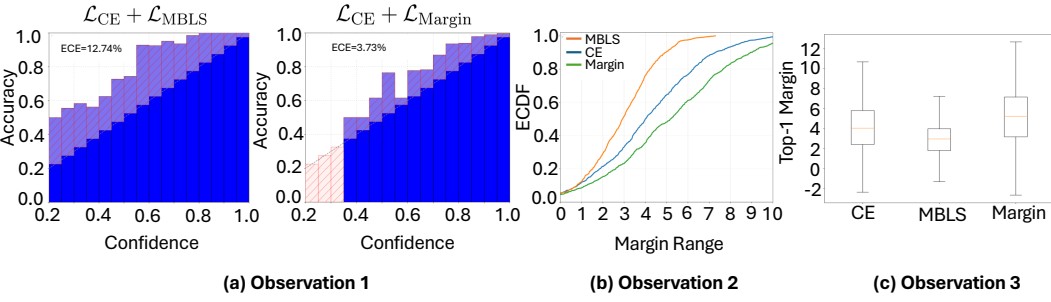

(a) Observation 1      (b) Observation 2      (c) Observation 3

Figure 5: **Margin based Label Smoothing(MBLS) vs Mean-Variance Margin(Margin) Regularization.** As shown in the base class reliability diagram (Observation 1), **MBLS** with Cross-Entropy (CE) shows underconfidence, while adding a **Margin** term alleviates this. For Observations 3 and 4, we train **CE**, **CE + MBLS**, and **CE + Margin** with MaPLe, compute the top-1 margin $m = z_y - \max_{j \neq y} z_j$, and plot the Empirical Cumulative Distribution Function (ECDF). The ECDF shows **Margin** yields fewer low-margin samples(Underconfident samples), **MBLS** trims large but leaves small ones, and **CE** lies in between. Box plots confirm this: **Margin** has the highest median ($\approx 5$) with a right-shifted IQR, **MBLS** caps extremes with a tight IQR, while **CE** retains a broad spread.

**Calibration Across Varying Shots:** Figure 4a shows classification and calibration metrics under different shot counts (4, 8, 16, 32) for both base and novel classes. Our method maintains low ECE across different prompt tuning settings, demonstrating robust calibration even in extreme few-shot scenarios

**Robustness to Prompt Initialization:** Figure 4b evaluates the robustness of our method to different prompt initialization strategies, averaged over 10 datasets, comparing in between vanilla **Maple** Khattak et al. (2023b) and **Maple+Ours**. **Maple+Ours** demonstrates consistent performance across different initialization choices (see suppl.) for both base and novel classes. This robustness across initialization schemes suggests that our regularization framework addresses the fundamental geometric factors underlying miscalibration rather than optimizing for specific conditions.

**Margin vs MBLS:** Compared to MBLS, Margin regularization reduces underconfidence by lifting low-margin samples (Figure 5, Observations 1–3), whereas MBLS primarily caps extreme margins.

## 5 CONCLUSION

We proposed a method to calibrate prompt-tuned VLMs by addressing a key challenge: preserving predictive reliability without distorting CLIP's semantic structure. Our approach introduces two complementary regularizers, a mean-variance logit margin loss and a moment-matching constraint on text embeddings, to jointly enforce geometric fidelity and calibration. Our framework requires no architectural changes and integrates seamlessly with existing prompt tuning setups, making it broadly applicable across domains. Evaluated on 11 datasets and multiple prompt-tuning strategies, our method consistently improves calibration, particularly on novel classes, without compromising classification accuracy. By decoupling predictive uncertainty from semantic drift, it enables more trustworthy and robust deployment of VLMs in real-world, open-vocabulary scenarios. We hope this work encourages further research into calibration-aware adaptations of foundation models.

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

# A APPENDIX

In the Appendix material, we provide the following:

- **Calibration analysis for base and novel classes across prompt learning methods** (Sec. A.1)
- **Robustness evaluation under natural distribution shifts** (Sec. A.2)
- **Additional results for ACE and MCE performance metrics** (Sec. A.3)
- **Hyperparameters details** (Sec. A.4)
- **Prompt templates and variations** (Sec. A.5)
- **Variance analysis** (Sec. A.6)
- **Results on Different Backbones**(Sec. A.7)
- **Decision Boundary Visualization**(Sec. A.8)
- **Reproducibility statement**(Sec. A.9)
- **The Use of Large Language Models (LLMs)**(Sec. A.10)
- **Limitations**(Sec. A.11)

## A.1 CALIBRATION ANALYSIS FOR BASE AND NOVEL CLASSES ACROSS PROMPT LEARNING METHODS

In the main paper, we provide the calibration results for three representative prompt learning methods: CoOp Zhou et al. (2022b), KgCoOp Yao et al. (2023a), and MaPLe Khattak et al. (2023b). Here we provide the results for additional prompt learning methods including CoCoOp Zhou et al. (2022a), ProDA Yao et al. (2023b), ProGrad Zhu et al. (2023), and PromptSRC Khattak et al. (2023c) for both base and novel classes to demonstrate the broader applicability and effectiveness of our calibration approach.

**Calibration Performance on Base Classes.** Table 4 presents the accuracy and calibration performance on base classes across 10 fine-grained classification benchmarks. The results consistently demonstrate that our calibration approach achieves superior calibration performance (lower ECE) while maintaining competitive accuracy across all evaluated prompt learning methods. Notably, our method shows the most significant improvements with PromptSRC Khattak et al. (2023c), reducing the average ECE from 4.26 to 3.04 without compromising clean accuracy. The consistent improvements across diverse prompt learning architectures validate the generalizability of our approach.

**Calibration Performance on Novel Classess.** We further evaluate the calibration performance on novel classes to assess the generalization capability of our approach when dealing with unseen categories during training. Table 5 presents the accuracy and calibration performance on novel classes across the same 10 benchmarks. Our method demonstrates consistent calibration improvements across all prompt learning methods when evaluated on novel classes. The results show that our approach effectively generalizes to unseen categories, with particularly notable improvements in ECE reduction. For instance, with ProDA Yao et al. (2023b), our method reduces the average ECE from 9.03 to 3.42 while maintaining comparable clean accuracy (70.22 vs 71.38). Similarly, with CoCoOp Zhou et al. (2022a), we achieve an ECE reduction from 5.55 to 3.86. These results highlight the robustness of our calibration approach in handling the challenging scenario of novel class prediction, where models are more prone to overconfidence due to limited training exposure.

## A.2 RESULTS ON NATURAL DISTRIBUTION SHIFTS

Here, we provide results on out-of-distribution datasets for the ImageNet-A Hendrycks et al. (2021b), ImageNet-V2 Shankar et al. (2020), ImageNet-R Hendrycks et al. (2021a), and ImageNet-S Wang et al. (2019) datasets demonstrating the robustness of our calibration approach under natural distribution shifts.

**Base Classes.** Table 6 presents the accuracy and calibration performance on base classes across these 4 natural distribution shift datasets. Our method consistently outperforms baseline calibration

Table 4: **Accuracy and calibration performance on base classes across 10 fine-grained classification benchmarks**. We report top-1 accuracy (Acc) and Expected Calibration Error (ECE) for multiple prompt-tuning strategies and diverse calibration baselines. Higher Acc. indicates better classification performance, while lower ECE reflects better calibration.

| Method | | Calt | Pets | Cars | Flow | Food | Air | SUN | DTD | Euro | UCF | Avg |
|---|---|---|---|---|---|---|---|---|---|---|---|---|
| Zero Shot | Acc. | 97.20 | 91.30 | 63.60 | 71.80 | 90.10 | 27.70 | 69.40 | 53.00 | 57.00 | 71.00 | 69.50 |
| | ECE | 6.49 | 2.25 | 3.74 | 3.11 | 1.57 | 3.03 | 1.59 | 4.53 | 8.35 | 3.24 | 3.58 |
| **CoCoOp** Zhou et al. (2022a) | | | | | | | | | | | | |
| CoCoOp Zhou et al. (2022a) | Acc. | 97.77 | 95.02 | 70.72 | 94.62 | 90.43 | 35.33 | 79.19 | 75.54 | 85.40 | 81.71 | 80.57 |
| | ECE | 1.43 | 3.21 | 6.89 | 7.85 | 0.86 | 5.42 | 3.78 | 3.88 | 8.09 | 3.78 | 4.52 |
| ZS-Norm Murugesan et al. (2024) | Acc. | 97.83 | 95.22 | 70.65 | 95.00 | 90.63 | 36.03 | 79.70 | 76.35 | 82.53 | 81.90 | 80.58 |
| | ECE | 2.93 | 3.20 | 8.67 | 7.56 | 1.49 | 8.50 | 7.50 | 10.82 | 16.09 | 4.08 | 7.08 |
| Penalty Murugesan et al. (2024) | Acc. | 97.83 | 94.98 | 69.95 | 92.43 | 90.72 | 34.35 | 79.34 | 71.49 | 69.57 | 80.49 | 78.12 |
| | ECE | 5.00 | 6.06 | 10.36 | 14.90 | 3.95 | 6.83 | 6.69 | 17.22 | 20.94 | 7.17 | 9.91 |
| **Ours** | Acc. | 97.93 | 94.69 | 69.42 | 94.11 | 90.51 | 34.25 | 78.92 | 74.77 | 84.70 | 82.34 | 80.16 |
| | ECE | 1.13 | 2.31 | 7.01 | 7.98 | 0.49 | 5.81 | 2.63 | 3.70 | 6.47 | 3.23 | 4.08 |
| **ProDA** Yao et al. (2023b) | | | | | | | | | | | | |
| ProDA Yao et al. (2023b) | Acc. | 97.61 | 94.75 | 69.76 | 89.96 | 89.33 | 33.01 | 76.17 | 70.02 | 81.83 | 79.99 | 78.24 |
| | ECE | 1.06 | 1.67 | 3.86 | 6.07 | 0.86 | 3.52 | 6.66 | 10.25 | 3.73 | 2.56 | 4.02 |
| ZS-Norm Murugesan et al. (2024) | Acc. | 97.55 | 94.37 | 69.77 | 89.62 | 89.50 | 33.03 | 76.46 | 71.33 | 82.00 | 79.33 | 78.30 |
| | ECE | 1.93 | 2.22 | 4.74 | 7.41 | 1.31 | 3.00 | 6.62 | 3.73 | 11.93 | 2.72 | 4.56 |
| Penalty Murugesan et al. (2024) | Acc. | 97.35 | 94.61 | 69.32 | 89.14 | 90.36 | 32.17 | 76.94 | 59.80 | 63.86 | 78.07 | 75.16 |
| | ECE | 4.00 | 8.11 | 7.86 | 12.89 | 3.79 | 4.91 | 2.09 | 11.28 | 18.05 | 7.83 | 8.08 |
| **Ours** | Acc. | 97.20 | 94.31 | 69.50 | 87.88 | 90.02 | 32.99 | 76.96 | 72.91 | 82.70 | 80.63 | 78.51 |
| | ECE | 1.75 | 2.29 | 6.91 | 8.21 | 1.18 | 3.38 | 1.18 | 2.27 | 5.37 | 2.50 | 3.50 |
| **ProGrad** Zhu et al. (2023) | | | | | | | | | | | | |
| ProGrad Zhu et al. (2023) | Acc. | 97.72 | 94.67 | 69.29 | 81.26 | 90.33 | 31.35 | 76.88 | 67.13 | 79.27 | 78.20 | 76.61 |
| | ECE | 3.53 | 3.83 | 6.84 | 6.82 | 1.65 | 2.60 | 3.70 | 6.38 | 12.24 | 3.92 | 5.15 |
| ZS-Norm Murugesan et al. (2024) | Acc. | 97.87 | 94.36 | 69.27 | 82.91 | 90.45 | 32.35 | 77.92 | 70.37 | 75.57 | 78.70 | 76.98 |
| | ECE | 5.51 | 4.85 | 10.02 | 10.95 | 2.67 | 8.50 | 10.33 | 23.04 | 17.46 | 6.47 | 9.98 |
| Penalty Murugesan et al. (2024) | Acc. | 97.81 | 94.21 | 69.02 | 84.14 | 90.47 | 32.65 | 77.33 | 57.29 | 67.21 | 75.65 | 74.58 |
| | ECE | 5.51 | 6.39 | 8.69 | 13.38 | 3.49 | 6.70 | 5.54 | 7.70 | 16.49 | 5.79 | 7.97 |
| **Ours** | Acc. | 97.55 | 94.45 | 68.95 | 82.62 | 90.29 | 31.57 | 77.03 | 68.52 | 79.78 | 79.04 | 76.98 |
| | ECE | 3.18 | 3.46 | 7.01 | 6.90 | 1.36 | 3.06 | 3.10 | 6.41 | 11.20 | 3.12 | 4.88 |
| **PromptSRC** Khattak et al. (2023c) | | | | | | | | | | | | |
| PromptSRC Khattak et al. (2023c) | Acc. | 98.08 | 95.36 | 78.15 | 97.95 | 90.60 | 40.74 | 82.63 | 83.41 | 93.17 | 87.09 | 84.72 |
| | ECE | 2.31 | 2.64 | 8.65 | 5.15 | 1.17 | 5.26 | 2.75 | 2.56 | 9.27 | 2.84 | 4.26 |
| ZS-Norm Murugesan et al. (2024) | Acc. | 98.21 | 95.43 | 77.55 | 97.50 | 90.78 | 40.78 | 82.62 | 81.56 | 48.78 | 85.76 | 79.90 |
| | ECE | 4.41 | 4.58 | 12.01 | 8.65 | 3.21 | 10.37 | 5.43 | 20.69 | 20.10 | 6.73 | 9.62 |
| Penalty Murugesan et al. (2024) | Acc. | 98.01 | 95.69 | 77.45 | 97.82 | 90.36 | 40.89 | 82.29 | 81.84 | 48.38 | 85.91 | 79.86 |
| | ECE | 5.41 | 4.78 | 10.61 | 9.65 | 4.51 | 12.47 | 6.43 | 18.69 | 19.10 | 8.98 | 10.06 |
| **Ours** | Acc. | 98.30 | 95.57 | 79.11 | 98.39 | 90.67 | 42.50 | 82.77 | 83.83 | 95.05 | 86.87 | 85.31 |
| | ECE | 1.03 | 0.89 | 8.96 | 0.97 | 0.90 | 6.04 | 1.34 | 4.59 | 4.33 | 1.39 | 3.04 |

approaches across all datasets. Notably, our approach achieves superior calibration with an average ECE of 2.82 compared to the vanilla MaPLe baseline (3.19), ZS-Norm (5.41), and Penalty (5.76). The improvements are particularly pronounced on challenging datasets like ImageNet-A and ImageNet-R, where our method reduces ECE from 2.52 to 2.21 and from 3.14 to 2.05, respectively, while maintaining competitive accuracy.

**Novel Classes.** Table 7 shows the corresponding results on novel classes under distribution shift. The consistent performance across both base and novel classes demonstrates the generalization capability of our calibration approach. Our method achieves an average ECE of 2.75 on novel classes, significantly outperforming ZS-Norm (5.31) and Penalty (5.64) baselines. The robustness across different types of distribution shifts, including adversarial examples (ImageNet-A), renditions (ImageNet-R), and sketch-like images (ImageNet-S), validates that our approach addresses fundamental calibration issues rather than dataset-specific artifacts.

These results are particularly important for real-world deployment scenarios where models encounter data that differs from the training distribution. The consistent calibration improvements across diverse distribution shifts demonstrate that our method provides reliable confidence estimates even under challenging out-of-distribution conditions.

Table 5: **Accuracy and calibration performance on novel classes across 10 fine-grained classification benchmarks**. We report top-1 accuracy (Acc) and Expected Calibration Error (ECE) for multiple prompt-tuning strategies and diverse calibration baselines. Higher Acc. indicates better classification performance, while lower ECE reflects better calibration.

| Method | | Calt | Pets | Cars | Flow | Food | Air | SUN | DTD | Euro | UCF | Avg |
|---|---|---|---|---|---|---|---|---|---|---|---|---|
| Zero Shot | Acc. | 94.10 | 97.10 | 75.00 | 77.50 | 91.10 | 35.90 | 75.50 | 60.60 | 63.80 | 78.60 | 74.30 |
| | ECE | 1.60 | 3.42 | 3.31 | 4.91 | 1.83 | 6.55 | 3.48 | 6.86 | 9.12 | 5.52 | 4.43 |
| **CoCoOp** Zhou et al. (2022a) | | | | | | | | | | | | |
| CoCoOp Zhou et al. (2022a) | Acc. | 94.51 | 97.69 | 73.26 | 72.27 | 91.02 | 33.47 | 76.54 | 57.65 | 63.14 | 74.89 | 73.44 |
| | ECE | 1.84 | 2.64 | 1.88 | 9.17 | 1.64 | 10.93 | 2.21 | 11.26 | 9.06 | 4.90 | 5.55 |
| ZS-Norm Murugesan et al. (2024) | Acc. | 94.76 | 97.24 | 73.56 | 70.45 | 91.43 | 32.97 | 76.84 | 54.59 | 57.01 | 71.64 | 72.05 |
| | ECE | 2.63 | 2.87 | 2.11 | 9.39 | 2.16 | 7.21 | 3.99 | 3.91 | 9.24 | 4.36 | 4.79 |
| Penalty Murugesan et al. (2024) | Acc. | 94.29 | 95.62 | 75.07 | 70.52 | 91.46 | 33.59 | 76.85 | 56.76 | 54.50 | 74.08 | 72.27 |
| | ECE | 2.22 | 5.11 | 5.69 | 5.80 | 4.22 | 5.30 | 3.93 | 11.13 | 11.44 | 3.66 | 5.85 |
| DAC | Acc. | - | - | - | - | - | - | - | - | - | - | - |
| | ECE | 3.65 | 2.43 | 2.21 | 7.74 | 1.64 | 9.03 | 1.09 | 7.47 | 13.49 | 2.70 | 5.15 |
| **Ours** | Acc. | 94.43 | 97.45 | 74.71 | 71.91 | 91.62 | 33.97 | 76.41 | 56.40 | 61.67 | 76.40 | 73.50 |
| | ECE | 1.51 | 2.58 | 3.22 | 6.36 | 0.96 | 8.85 | 0.87 | 4.77 | 6.68 | 2.76 | 3.86 |
| **ProDA** Yao et al. (2023b) | | | | | | | | | | | | |
| ProDA Yao et al. (2023b) | Acc. | 93.99 | 96.90 | 73.16 | 72.51 | 90.64 | 31.35 | 65.02 | 53.99 | 51.86 | 72.76 | 70.22 |
| | ECE | 3.22 | 1.96 | 3.18 | 8.51 | 0.84 | 15.03 | 14.08 | 16.90 | 21.85 | 4.74 | 9.03 |
| ZS-Norm Murugesan et al. (2024) | Acc. | 93.81 | 97.28 | 72.53 | 72.81 | 90.44 | 30.09 | 66.59 | 52.13 | 57.77 | 72.67 | 71.01 |
| | ECE | 2.36 | 2.42 | 2.06 | 8.34 | 0.94 | 10.76 | 12.12 | 7.65 | 8.75 | 4.41 | 5.98 |
| Penalty Murugesan et al. (2024) | Acc. | 93.92 | 97.20 | 73.39 | 73.57 | 90.70 | 32.45 | 67.73 | 50.48 | 60.05 | 72.49 | 71.20 |
| | ECE | 1.53 | 6.14 | 3.76 | 4.36 | 3.47 | 7.82 | 2.51 | 4.96 | 14.47 | 3.86 | 5.29 |
| DAC | Acc. | - | - | - | - | - | - | - | - | - | - | - |
| | ECE | 4.87 | 4.72 | 3.28 | 6.32 | 0.70 | 7.40 | 1.06 | 5.68 | 3.33 | 4.14 | 4.15 |
| **Ours** | Acc. | 93.56 | 97.56 | 73.81 | 72.74 | 91.14 | 30.57 | 66.18 | 53.82 | 58.58 | 75.79 | 71.38 |
| | ECE | 1.48 | 3.25 | 2.77 | 5.12 | 1.01 | 6.78 | 1.90 | 4.60 | 4.91 | 2.35 | 3.42 |
| **ProGrad** Zhu et al. (2023) | | | | | | | | | | | | |
| ProGrad Zhu et al. (2023) | Acc. | 94.76 | 97.32 | 74.85 | 75.29 | 91.06 | 34.43 | 75.42 | 56.44 | 61.98 | 78.74 | 74.03 |
| | ECE | 1.67 | 3.52 | 2.68 | 7.46 | 1.76 | 9.21 | 2.05 | 4.48 | 8.83 | 3.57 | 4.52 |
| ZS-Norm Murugesan et al. (2024) | Acc. | 94.43 | 97.37 | 74.97 | 75.18 | 91.18 | 31.49 | 74.79 | 55.80 | 67.97 | 77.39 | 74.06 |
| | ECE | 1.80 | 5.11 | 5.32 | 3.73 | 2.68 | 3.79 | 7.10 | 12.79 | 12.83 | 4.83 | 6.00 |
| Penalty Murugesan et al. (2024) | Acc. | 94.87 | 96.98 | 75.81 | 74.54 | 91.05 | 34.55 | 75.03 | 53.74 | 66.97 | 76.80 | 74.03 |
| | ECE | 1.90 | 5.54 | 5.07 | 4.80 | 3.08 | 5.31 | 3.08 | 4.96 | 14.86 | 5.05 | 5.37 |
| DAC | Acc. | - | - | - | - | - | - | - | - | - | - | - |
| | ECE | 1.97 | 3.31 | 2.29 | 5.04 | 1.85 | 10.46 | 1.32 | 3.49 | 6.90 | 2.42 | 3.91 |
| **Ours** | Acc. | 94.29 | 97.48 | 75.09 | 74.66 | 91.21 | 32.87 | 74.81 | 55.60 | 67.91 | 78.53 | 74.25 |
| | ECE | 1.03 | 3.26 | 1.98 | 5.14 | 1.47 | 9.34 | 2.32 | 3.30 | 6.06 | 3.06 | 3.70 |
| **PromptSRC** Khattak et al. (2023c) | | | | | | | | | | | | |
| PromptSRC Khattak et al. (2023c) | Acc. | 94.21 | 97.31 | 75.58 | 77.28 | 91.51 | 29.73 | 78.79 | 61.03 | 74.72 | 77.86 | 75.80 |
| | ECE | 1.51 | 3.26 | 2.06 | 5.50 | 1.77 | 12.92 | 1.07 | 6.68 | 8.08 | 2.81 | 4.57 |
| ZS-Norm Murugesan et al. (2024) | Acc. | 94.18 | 97.61 | 74.80 | 76.31 | 91.60 | 36.23 | 78.80 | 59.02 | 37.07 | 77.25 | 72.29 |
| | ECE | 2.06 | 4.30 | 3.63 | 5.86 | 3.66 | 4.64 | 3.57 | 12.29 | 12.56 | 3.95 | 5.65 |
| Penalty Murugesan et al. (2024) | Acc. | 94.17 | 97.55 | 74.12 | 76.34 | 91.99 | 36.63 | 78.14 | 59.62 | 37.37 | 77.76 | 72.37 |
| | ECE | 3.06 | 5.40 | 4.63 | 4.86 | 4.98 | 4.94 | 4.53 | 11.29 | 10.66 | 4.65 | 5.90 |
| DAC | Acc. | - | - | - | - | - | - | - | - | - | - | - |
| | ECE | 1.58 | 2.98 | 2.39 | 5.03 | 1.55 | 8.55 | 0.79 | 5.50 | 7.24 | 2.46 | 3.81 |
| **Ours** | Acc. | 94.29 | 97.28 | 74.49 | 75.44 | 91.68 | 36.85 | 78.39 | 57.53 | 72.64 | 77.72 | 75.63 |
| | ECE | 1.14 | 1.19 | 2.39 | 5.44 | 0.72 | 9.26 | 0.77 | 6.81 | 8.37 | 1.89 | 3.80 |

## A.3   ADDITIONAL RESULTS: ACE AND MCE PERFORMANCE METRICS

In the main paper, we evaluate classification performance using top-1 accuracy and model calibration using Expected Calibration Error (ECE). Here we provide comprehensive results for additional calibration metrics including Adaptive Calibration Error (ACE) Nixon et al. (2019) and Maximum Calibration Error (MCE) Naeini et al. (2015) to further validate the effectiveness of our approach.

Table 8 presents the MCE and ACE results on **base classes** across 10 fine-grained classification benchmarks. Our method demonstrates consistent improvements across both metrics for all evaluated prompt learning methods. For CoOp, our approach reduces the average MCE from 2.40 to 0.90 and ACE from 6.72 to 2.78, representing substantial calibration improvements. Similarly, with KGCoOp, we achieve reductions in MCE from 1.62 to 1.39 and ACE from 6.19 to 5.85. These results are particularly noteworthy as MCE captures the worst-case calibration error, indicating that our method not only improves average calibration but also reduces extreme miscalibration cases.

Table 6: **Accuracy and calibration performance on base classes across 4 natural distribution shift datasets**. We report top-1 accuracy (Acc) and Expected Calibration Error (ECE) for MaPLe Khattak et al. (2023b). Higher Acc. indicates better classification performance, while lower ECE reflects better calibration.

| Method | | INet-V2 | INet-S | INet-A | INet-R | Avg |
|---|---|---|---|---|---|---|
| **MaPLe** Khattak et al. (2023b) | | | | | | |
| MaPLe Khattak et al. (2023b) | Acc. | 67.35 | 53.35 | 68.31 | 85.22 | 68.56 |
| | ECE | 3.14 | 3.94 | 2.52 | 3.14 | 3.19 |
| ZS-Norm Murugesan et al. (2024) | Acc. | 66.15 | 53.24 | 68.41 | 85.17 | 68.49 |
| | ECE | 3.41 | 4.12 | 6.98 | 7.13 | 5.41 |
| Penalty Murugesan et al. (2024) | Acc. | 66.72 | 53.04 | 68.61 | 85.17 | 68.39 |
| | ECE | 3.61 | 4.62 | 7.48 | 7.33 | 5.76 |
| **Ours** | Acc. | 67.19 | 53.15 | 67.86 | 85.28 | 68.37 |
| | ECE | 3.09 | 3.91 | 2.21 | 2.05 | 2.82 |

Table 7: **Accuracy and calibration performance on novel classes across 4 natural distribution shift datasets**. We report top-1 accuracy (Acc) and Expected Calibration Error (ECE) for MaPLe Khattak et al. (2023b). Acc. indicates better classification performance, while lower ECE reflects better calibration.

| Method | | INet-V2 | INet-S | INet-A | INet-R | Avg |
|---|---|---|---|---|---|---|
| **MaPLe** Khattak et al. (2023b) | | | | | | |
| MaPLe Khattak et al. (2023b) | Acc. | 67.36 | 53.36 | 68.31 | 85.22 | 68.56 |
| | ECE | 3.16 | 3.73 | 2.52 | 3.14 | 3.14 |
| ZS-Norm Murugesan et al. (2024) | Acc. | 66.75 | 53.34 | 68.60 | 85.16 | 68.46 |
| | ECE | 3.54 | 4.01 | 6.47 | 7.21 | 5.31 |
| Penalty Murugesan et al. (2024) | Acc. | 66.75 | 53.11 | 68.60 | 85.16 | 68.41 |
| | ECE | 3.64 | 4.11 | 7.47 | 7.32 | 5.64 |
| **Ours** | Acc. | 67.20 | 53.21 | 67.86 | 85.28 | 68.39 |
| | ECE | 3.11 | 3.63 | 2.20 | 2.05 | 2.75 |

Table 9 shows the results on **novel classes**. The improvements are consistent with the base class results, demonstrating the generalization capability of our calibration approach. For KGCoOp on novel classes, our method maintains similar MCE performance (1.18 vs 1.17) while slightly improving ACE from 3.93 to 3.65. The robustness across different calibration metrics validates that our approach addresses fundamental calibration issues rather than optimizing for specific metrics.

The consistent improvements across ECE, MCE, and ACE metrics provide strong evidence that our calibration method effectively reduces both average and worst-case calibration errors, making it suitable for deployment in safety-critical applications.

### A.4 HYPERPARAMETERS DETAILS

For all experiments, we use CLIP (ViT-B/16) Radford et al. (2021) as the pre-trained vision-language model. Prompt-tuning is conducted in a few-shot setting with 16 samples per class, using a learning rate of 0.005 and a batch size of 8. For each baseline method, we adopt its official implementation and follow the recommended hyperparameter settings from the original papers. All experiments are performed on an NVIDIA RTX A6000 GPU with 48GB memory.

For our proposed calibration method, we use the following hyperparameters across all experiments: $\lambda_{\text{Margin}} = 1.0$ controls the strength of the margin-based regularization, $\alpha = 0.1$ balances the average marin, $\beta = 0.01$ is the weight for the variance loss, and $\lambda_{\text{mom}} = 5.0$ controls the local moment matching regularization. Table 10 and 11 show how we choose these values. These hyperparameters were fixed across all datasets and prompt learning methods to ensure fair comparison. We conduct 3 random seeds for each experiment and report the average results.

Table 8: **Calibration performance on base classes across 10 fine-grained classification benchmarks**. We report Maximum Calibration Error (MCE) and Adaptive Calibration Error (ACE) for multiple prompt-tuning strategies and diverse calibration baselines. Lower MCE and ACE reflects better calibration.

| Method | | Calt | Pets | Cars | Flow | Food | Air | SUN | DTD | Euro | UCF | Avg |
|---|---|---|---|---|---|---|---|---|---|---|---|---|
| **CoOp** Zhou et al. (2022b) | | | | | | | | | | | | |
| CoOp Zhou et al. (2022b) | MCE | 0.24 | 0.31 | 0.91 | 2.46 | 1.72 | 4.77 | 3.99 | 5.68 | 0.60 | 3.34 | 2.40 |
| | ACE | 0.44 | 0.62 | 3.65 | 4.67 | 3.65 | 25.70 | 8.11 | 12.01 | 1.95 | 6.41 | 6.72 |
| ZS-Norm Murugesan et al. (2024) | MCE | 1.40 | 2.17 | 2.20 | 2.76 | 0.90 | 4.34 | 0.82 | 14.76 | 11.82 | 0.87 | 4.20 |
| | ACE | 4.32 | 7.70 | 11.26 | 11.12 | 3.14 | 13.05 | 4.26 | 49.53 | 36.04 | 3.36 | 14.38 |
| Penalty Murugesan et al. (2024) | MCE | 1.62 | 1.85 | 2.13 | 2.50 | 1.38 | 2.23 | 0.94 | 4.30 | 11.34 | 1.63 | 2.99 |
| | ACE | 4.75 | 6.41 | 10.01 | 9.36 | 5.98 | 8.61 | 4.61 | 21.48 | 20.86 | 7.09 | 9.92 |
| **Ours** | MCE | 0.33 | 1.01 | 1.87 | 1.68 | 0.12 | 1.21 | 0.31 | 0.56 | 1.67 | 0.22 | 0.90 |
| | ACE | 0.98 | 2.10 | 7.55 | 4.94 | 0.21 | 2.40 | 1.30 | 2.01 | 5.12 | 1.15 | 2.78 |
| **MaPLe** Khattak et al. (2023b) | | | | | | | | | | | | |
| MaPLe Khattak et al. (2023b) | MCE | 0.51 | 0.60 | 1.68 | 1.29 | 0.34 | 0.91 | 0.19 | 1.04 | 0.74 | 0.56 | 0.79 |
| | ACE | 2.14 | 1.19 | 6.91 | 3.21 | 0.73 | 2.95 | 1.17 | 3.71 | 2.94 | 1.49 | 2.64 |
| ZS-Norm Murugesan et al. (2024) | MCE | 1.28 | 1.01 | 3.79 | 1.45 | 2.25 | 1.95 | 2.27 | 2.80 | 1.24 | 0.61 | 1.87 |
| | ACE | 5.28 | 3.21 | 20.73 | 7.21 | 11.17 | 6.98 | 8.59 | 12.37 | 6.91 | 3.40 | 8.59 |
| Penalty Murugesan et al. (2024) | MCE | 1.88 | 1.85 | 2.30 | 3.15 | 1.11 | 2.27 | 1.24 | 3.69 | 0.61 | 1.45 | 1.96 |
| | ACE | 5.68 | 6.29 | 11.17 | 11.97 | 3.71 | 8.72 | 6.81 | 20.23 | 3.10 | 7.61 | 8.53 |
| **Ours** | MCE | 0.62 | 0.34 | 1.20 | 0.87 | 1.47 | 0.62 | 1.11 | 1.50 | 0.60 | 0.94 | 0.93 |
| | ACE | 1.62 | 0.80 | 3.98 | 2.18 | 3.66 | 0.94 | 4.38 | 7.55 | 1.54 | 1.34 | 2.80 |
| **KGCoOp** Yao et al. (2023a) | | | | | | | | | | | | |
| KGCoOp Yao et al. (2023a) | MCE | 1.14 | 1.17 | 2.23 | 2.75 | 0.62 | 1.17 | 0.97 | 1.61 | 3.31 | 1.25 | 1.62 |
| | ACE | 2.56 | 2.95 | 10.14 | 11.95 | 1.59 | 2.95 | 4.91 | 8.39 | 11.90 | 4.59 | 6.19 |
| ZS-Norm Murugesan et al. (2024) | MCE | 1.31 | 1.13 | 2.28 | 3.02 | 0.64 | 3.12 | 1.34 | 3.85 | 4.06 | 1.12 | 2.19 |
| | ACE | 2.98 | 3.06 | 10.58 | 13.00 | 1.69 | 9.59 | 6.51 | 20.40 | 15.58 | 5.75 | 8.91 |
| Penalty Murugesan et al. (2024) | MCE | 1.27 | 1.40 | 2.10 | 3.03 | 0.81 | 1.67 | 1.20 | 2.87 | 3.30 | 1.33 | 1.90 |
| | ACE | 4.20 | 4.61 | 9.96 | 12.53 | 2.44 | 6.41 | 5.92 | 10.66 | 13.14 | 6.04 | 7.59 |
| **Ours** | MCE | 0.86 | 1.05 | 1.12 | 2.01 | 0.55 | 1.94 | 0.81 | 1.12 | 3.61 | 0.86 | 1.39 |
| | ACE | 1.83 | 2.83 | 8.1 | 11.21 | 1.42 | 5.11 | 4.15 | 7.21 | 12.65 | 3.95 | 5.85 |

## A.5 PROMPT TEMPLATES AND VARIATIONS

In the main paper, Figure 4b presents our method's robustness to different prompt initialization. The following prompt templates were evaluated to assess initialization robustness: "a nice image of a {}", "an example of a {}", "a picture of a {}", and "a photo of the cool {}". These templates represent different stylistic and semantic variations commonly used in prompt learning literature. This robustness is particularly valuable in practical deployment scenarios where optimal prompt initialization may not be known in advance.

## A.6 VARIANCE ANALYSIS

To assess the statistical robustness of our approach, we evaluate the variance in performance across 3 random seeds for both accuracy and calibration metrics. Table 12 presents the variance results across 9 novel classes of fine-grained classification benchmarks for CoCoOp and KGCoOp methods. Our approach demonstrates superior stability with consistently lower variance in both accuracy and ECE compared to baseline calibration methods. For CoCoOp, our method achieves significantly lower average variance in accuracy (1.25 vs 1.55) and ECE (0.55 vs 2.53) compared to the vanilla baseline. Similarly, with KGCoOp, we maintain competitive variance performance with average accuracy variance of 0.44 compared to the baseline's 0.57, while substantially reducing ECE variance from 0.58 to 0.21. The reduced variance in calibration error is particularly noteworthy as it indicates that our method provides more consistent and reliable confidence estimates across different experimental runs, which is crucial for deployment in safety-critical applications.

## A.7 RESULTS ON DIFFERENT BACKBONES

To evaluate the adaptability of our method, we conduct experiments on CoOp Zhou et al. (2022b) with different backbones, namely RN-50 and ViT-B/32. The Tables 13 and 14 results show that our approach consistently outperforms existing methods across both backbones, while also maintaining

Table 9: **Calibration performance on novel classes across 10 fine-grained classification benchmarks**. We report Maximum Calibration Error (MCE) and Adaptive Calibration Error (ACE) for multiple prompt-tuning strategies and diverse calibration baselines. Lower MCE and ACE reflects better calibration.

| Method | | Calt | Pets | Cars | Flow | Food | Air | SUN | DTD | Euro | UCF | Avg |
|---|---|---|---|---|---|---|---|---|---|---|---|---|
| **CoOp** Zhou et al. (2022b) | | | | | | | | | | | | |
| CoOp Zhou et al. (2022b) | MCE | 2.61 | 0.64 | 2.38 | 5.48 | 1.44 | 5.01 | 4.35 | 7.06 | 4.41 | 6.11 | 3.95 |
| | ACE | 3.47 | 1.67 | 12.45 | 18.33 | 3.84 | 28.41 | 13.92 | 26.88 | 12.73 | 19.17 | 14.09 |
| DAC Wang et al. (2024a) | MCE | 1.50 | 0.66 | 1.21 | 2.08 | 0.49 | 3.57 | 1.03 | 2.24 | 3.03 | 1.70 | 1.76 |
| | ACE | 2.60 | 1.70 | 5.17 | 10.18 | 1.75 | 17.27 | 4.00 | 10.51 | 8.58 | 8.63 | 7.04 |
| ZS-Norm Murugesan et al. (2024) | MCE | 1.46 | 2.06 | 0.75 | 1.08 | 0.82 | 2.24 | 0.55 | 9.32 | 11.82 | 1.14 | 3.12 |
| | ACE | 2.59 | 7.9 | 3.01 | 5.93 | 3.3 | 9.86 | 2.27 | 21.97 | 37.04 | 3.91 | 9.78 |
| Penalty Murugesan et al. (2024) | MCE | 0.92 | 2.1 | 0.68 | 1.16 | 1.00 | 2.46 | 0.76 | 0.9 | 7.54 | 1.36 | 1.89 |
| | ACE | 2.23 | 7.32 | 2.69 | 5.46 | 4.69 | 7.44 | 2.91 | 4.35 | 14.94 | 4.51 | 5.65 |
| **Ours** | MCE | 1.05 | 1.26 | 0.55 | 0.98 | 0.25 | 2.9 | 0.64 | 1.97 | 3.9 | 1.49 | 1.42 |
| | ACE | 2.3 | 2.92 | 2.02 | 3.67 | 0.91 | 10.47 | 2.99 | 9.6 | 11.24 | 5.04 | 4.83 |
| **MaPLe** Khattak et al. (2023b) | | | | | | | | | | | | |
| MaPLe Khattak et al. (2023b) | MCE | 0.55 | 1.04 | 0.56 | 5.02 | 0.37 | 1.48 | 0.65 | 3.50 | 1.66 | 0.66 | 1.55 |
| | ACE | 1.26 | 2.44 | 2.83 | 12.67 | 1.12 | 7.27 | 2.49 | 14.90 | 7.90 | 2.81 | 5.57 |
| DAC Wang et al. (2024a) | MCE | 0.39 | 1.06 | 0.77 | 4.42 | 0.50 | 2.26 | 0.40 | 1.95 | 2.55 | 0.66 | 1.50 |
| | ACE | 1.19 | 2.44 | 2.57 | 11.28 | 1.48 | 8.90 | 1.37 | 8.24 | 9.12 | 2.32 | 4.89 |
| ZS-Norm Murugesan et al. (2024) | MCE | 0.84 | 1.23 | 2.13 | 1.76 | 1.32 | 1.92 | 1.07 | 1.40 | 0.91 | 7.51 | 2.01 |
| | ACE | 1.62 | 4.15 | 9.46 | 7.77 | 3.83 | 7.85 | 3.77 | 6.11 | 4.84 | 13.73 | 6.31 |
| Penalty Murugesan et al. (2024) | MCE | 0.84 | 1.72 | 1.40 | 1.32 | 1.11 | 1.07 | 0.91 | 2.73 | 7.11 | 1.76 | 2.00 |
| | ACE | 1.92 | 7.60 | 6.41 | 4.83 | 4.05 | 3.47 | 4.74 | 10.46 | 16.73 | 8.67 | 6.89 |
| **Ours** | MCE | 0.44 | 0.52 | 1.61 | 1.08 | 1.70 | 0.88 | 2.66 | 0.55 | 0.14 | 0.87 | 1.05 |
| | ACE | 0.32 | 0.95 | 6.38 | 3.13 | 11.27 | 2.07 | 8.49 | 2.48 | 0.58 | 4.74 | 4.04 |
| **KGCoOp** Yao et al. (2023a) | | | | | | | | | | | | |
| KGCoOp Yao et al. (2023a) | MCE | 0.53 | 1.16 | 0.9 | 1.06 | 0.74 | 1.78 | 0.39 | 1.22 | 3.37 | 0.69 | 1.18 |
| | ACE | 1.22 | 3.26 | 3.36 | 5.45 | 2.01 | 5.86 | 1.83 | 5.02 | 8.69 | 2.59 | 3.93 |
| DAC Wang et al. (2024a) | MCE | 0.62 | 1.21 | 0.85 | 1.37 | 0.64 | 3.02 | 0.42 | 1.33 | 1.75 | 0.77 | 1.20 |
| | ACE | 1.56 | 3.02 | 3.11 | 6.61 | 1.93 | 11.74 | 1.82 | 7.26 | 6.63 | 2.70 | 4.64 |
| ZS-Norm Murugesan et al. (2024) | MCE | 0.58 | 1.17 | 0.94 | 1.06 | 0.75 | 2.97 | 0.59 | 1.90 | 1.52 | 0.97 | 1.25 |
| | ACE | 1.33 | 3.30 | 3.86 | 5.31 | 2.10 | 8.39 | 3.30 | 5.95 | 6.51 | 3.81 | 4.39 |
| Penalty Murugesan et al. (2024) | MCE | 0.56 | 1.43 | 0.99 | 1.38 | 0.87 | 1.34 | 0.58 | 1.68 | 3.72 | 1.28 | 1.38 |
| | ACE | 1.19 | 3.51 | 3.89 | 5.14 | 2.64 | 5.00 | 3.35 | 5.63 | 13.86 | 4.23 | 4.84 |
| **Ours** | MCE | 0.51 | 1.21 | 0.85 | 0.98 | 0.68 | 2.78 | 0.47 | 1.97 | 1.35 | 0.91 | 1.17 |
| | ACE | 1.1 | 3.43 | 3.68 | 4.91 | 1.92 | 7.85 | 2.01 | 4.11 | 4.32 | 3.15 | 3.65 |

Table 10: Hyperparameter search for $\alpha$ (0.1–0.3) and $\beta$ (0.01 and 0.05), with results averaged across Caltech Fei-Fei et al. (2004), Food101 Bossard et al. (2014), and DTD Cimpoi et al. (2014), reported on both base and novel classes.

| $(\alpha, \beta)$ | ACC | | | ECE | | |
|---|---|---|---|---|---|---|
| | Base | Novel | Avg | Base | Novel | Avg |
| (0.1, 0.01) | 89.65 | 82.38 | 86.02 | 1.99 | 5.66 | 3.83 |
| (0.2, 0.01) | 89.72 | 79.66 | 84.69 | 3.46 | 8.53 | 5.99 |
| (0.3, 0.01) | 89.28 | 80.72 | 85.00 | 4.46 | 9.50 | 6.98 |
| (0.1, 0.05) | 89.12 | 79.99 | 84.56 | 3.95 | 6.26 | 5.10 |
| (0.2, 0.05) | 83.39 | 81.49 | 82.44 | 4.10 | 8.20 | 6.15 |
| (0.3, 0.05) | 89.75 | 80.52 | 85.14 | 2.01 | 8.01 | 5.01 |

improvements in accuracy. In base classes, for RN-50, our method achieves an average ECE of 3.46 compared to 4.04 for the vanilla baseline. Similarly, for ViT-B/32, our method attains an ECE of 2.87, outperforming the vanilla baseline at 3.15. For novel classes, our method achieves the second-lowest average ECE of 5.46 on RN-50, with ZS-Norm Murugesan et al. (2024) performing slightly better at 5.23. In contrast, on ViT-B/32, our method achieves the lowest ECE of 5.82, surpassing all other approaches.

Table 11: Hyperparameter search for $\lambda_{\text{mom}}$ over values 1–10, with results averaged across Caltech Fei-Fei et al. (2004), Food101 Bossard et al. (2014), and DTD Cimpoi et al. (2014), reported on both base and novel classes.

| | ACC | | | ECE | | |
|---|---|---|---|---|---|---|
| $\lambda$ | Base | Novel | Avg | Base | Novel | Avg |
| 1 | 89.44 | 81.51 | 85.48 | 3.47 | 4.54 | 4.01 |
| 3 | 89.17 | 82.14 | 85.65 | 2.11 | 3.94 | 3.03 |
| 5 | 89.71 | 82.64 | 86.18 | 1.80 | 3.22 | 2.51 |
| 8 | 89.93 | 82.44 | 86.19 | 2.99 | 5.24 | 4.12 |
| 10 | 89.15 | 82.00 | 85.58 | 3.64 | 5.21 | 4.43 |

Table 12: **Variance across 3 random seeds for 9 novel classes of fine-grained classification benchmarks**.

| Method | | Calt | Pets | Cars | Flow | Food | Air | SUN | DTD | Euro | **Avg** |
|---|---|---|---|---|---|---|---|---|---|---|---|
| **CoCoOp** Zhou et al. (2022b) | | | | | | | | | | | |
| CoCoOp Zhou et al. (2022b) | Var. Acc | 0.81 | 0.08 | 2.65 | 1.53 | 4.41 | 0.01 | 3.61 | 0.19 | 0.69 | 1.55 |
| | Var. ECE | 0.36 | 0.03 | 3.09 | 8.82 | 3.88 | 0.01 | 6.25 | 0.10 | 0.21 | 2.53 |
| ZS-Norm Murugesan et al. (2024) | Var. Acc | 1.02 | 0.24 | 4.60 | 4.62 | 0.58 | 0.12 | 1.29 | 0.19 | 0.33 | 1.44 |
| | Var. ECE | 1.69 | 0.12 | 0.88 | 6.66 | 4.45 | 0.10 | 1.04 | 0.23 | 5.11 | 2.25 |
| Penalty Murugesan et al. (2024) | Var. Acc | 0.12 | 0.09 | 0.11 | 0.32 | 0.59 | 5.81 | 2.25 | 0.09 | 0.08 | 1.05 |
| | Var. ECE | 0.08 | 0.02 | 0.20 | 0.04 | 0.75 | 1.00 | 2.50 | 0.19 | 0.02 | 0.53 |
| **Ours** | Var. Acc | 0.05 | 0.01 | 7.72 | 1.21 | 1.96 | 0.07 | 0.01 | 0.18 | 0.01 | 1.25 |
| | Var. ECE | 0.01 | 0.03 | 2.43 | 0.01 | 1.46 | 0.00 | 0.73 | 0.21 | 0.06 | 0.55 |
| **KGCoOp** Yao et al. (2023a) | | | | | | | | | | | |
| KGCoOp Yao et al. (2023a) | Var. Acc | 0.03 | 0.00 | 2.19 | 1.21 | 0.58 | 0.01 | 0.85 | 0.15 | 0.08 | 0.57 |
| | Var. ECE | 0.04 | 0.00 | 4.00 | 0.25 | 0.72 | 0.00 | 0.01 | 0.01 | 0.17 | 0.58 |
| ZS-Norm Murugesan et al. (2024) | Var. Acc | 0.05 | 0.00 | 7.29 | 0.92 | 0.50 | 0.01 | 3.35 | 0.35 | 0.00 | 1.39 |
| | Var. ECE | 0.04 | 0.00 | 2.31 | 0.07 | 0.01 | 0.01 | 1.80 | 0.00 | 0.04 | 0.48 |
| Penalty Murugesan et al. (2024) | Var. Acc | 0.01 | 0.00 | 1.66 | 0.16 | 1.23 | 0.01 | 1.04 | 0.11 | 0.15 | 0.49 |
| | Var. ECE | 0.08 | 0.06 | 0.38 | 0.18 | 1.19 | 0.01 | 2.37 | 0.07 | 0.06 | 0.49 |
| **Ours** | Var. Acc | 0.05 | 0.00 | 2.31 | 0.17 | 0.96 | 0.02 | 0.00 | 0.40 | 0.04 | 0.44 |
| | Var. ECE | 0.02 | 0.00 | 0.16 | 0.74 | 0.44 | 0.04 | 0.02 | 0.10 | 0.36 | 0.21 |

## A.8 DECISION BOUNDARY VISUALIZATION

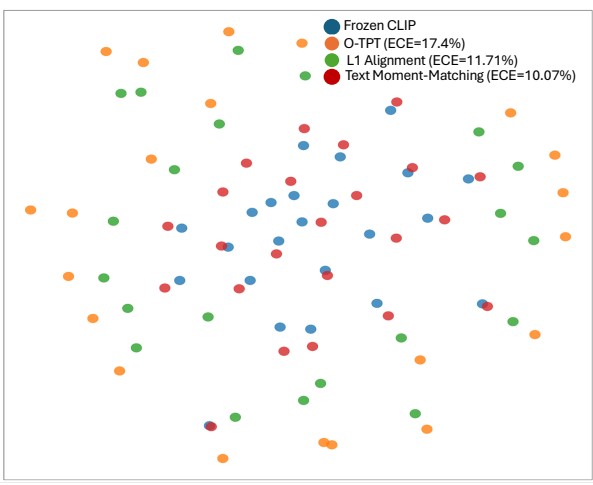

Figure 6: T-SNE visualization of Text Features

Table 13: **Accuracy and calibration performance on base classes across 10 fine-grained classification benchmarks using RN-50 and ViT-B/32**. We report top-1 accuracy (Acc) and Expected Calibration Error (ECE) for CoOpZhou et al. (2022b), a prompt-tuning strategy, evaluated with different backbones.

| Method | | Calt | Pets | Cars | Flow | Food | Air | SUN | DTD | Euro | UCF | Avg |
|---|---|---|---|---|---|---|---|---|---|---|---|---|
| **CoOp-RN50** Zhou et al. (2022b) | | | | | | | | | | | | |
| CoOp-RN50 Zhou et al. (2022b) | Acc. | 95.22 | 90.5 | 70.22 | 95.25 | 82.54 | 29.95 | 76.37 | 74.85 | 81.32 | 80.4 | 76.99 |
| | ECE | 1.27 | 2.32 | 6.15 | 4.22 | 1.15 | 2.62 | 2.81 | 8.65 | 9.34 | 1.82 | 4.04 |
| ZS-Norm Murugesan et al. (2024) | Acc. | 95.55 | 90.96 | 69.89 | 95.19 | 82.71 | 25.89 | 76.67 | 74.27 | 89.57 | 79.58 | 78.03 |
| | ECE | 4.04 | 5.63 | 10.52 | 9.92 | 3.4 | 17.28 | 4.35 | 37.15 | 35.76 | 7.18 | 13.52 |
| Penalty Murugesan et al. (2024) | Acc. | 96.31 | 92.1 | 68.58 | 94.65 | 83.66 | 26.63 | 74.57 | 68.48 | 47.61 | 77.9 | 78.12 |
| | ECE | 6.21 | 8.69 | 11.85 | 11.34 | 5.29 | 6.4 | 5.68 | 18.93 | 22.61 | 9.95 | 10.71 |
| **Ours** | Acc. | 95.44 | 91.08 | 69.90 | 95.22 | 82.77 | 29.01 | 77.07 | 75.42 | 89.79 | 79.96 | 78.57 |
| | ECE | 1.37 | 2.21 | 8.65 | 4.74 | 0.84 | 3.32 | 1.23 | 5.87 | 4.12 | 2.21 | 3.46 |
| **CoOp–ViT-B/32** Zhou et al. (2022b) | | | | | | | | | | | | |
| CoOp–ViT-B/32 Zhou et al. (2022b) | Acc. | 97.05 | 92.71 | 73.13 | 95.28 | 84.93 | 31.87 | 79.16 | 77.55 | 91.10 | 82.83 | 79.88 |
| | ECE | 1.17 | 2.45 | 4.86 | 4.18 | 1.06 | 3.12 | 2.5 | 6.15 | 3.91 | 2.09 | 3.15 |
| ZS-Norm Murugesan et al. (2024) | Acc. | 97.18 | 92.08 | 72.86 | 94.81 | 85.12 | 32.19 | 79.64 | 76.23 | 90.09 | 82.76 | 80.25 |
| | ECE | 1.98 | 7.96 | 9.70 | 7.91 | 6.00 | 12.97 | 3.48 | 38.62 | 39.31 | 5.25 | 13.32 |
| Penalty Murugesan et al. (2024) | Acc. | 96.73 | 93.22 | 73.33 | 94.75 | 86.05 | 29.99 | 78.95 | 69.64 | 53.42 | 5.94 | 75.78 |
| | ECE | 4.88 | 6.66 | 10.54 | 10.05 | 4.45 | 4.82 | 5.42 | 22.00 | 22.15 | 5.94 | 9.69 |
| **Ours** | Acc. | 97.46 | 93.14 | 73.78 | 95.09 | 85.12 | 31.83 | 79.58 | 79.01 | 91.09 | 82.87 | 80.09 |
| | ECE | 0.92 | 1.68 | 6.77 | 4.53 | 0.79 | 2.91 | 0.95 | 4.77 | 3.52 | 1.87 | 2.87 |

Table 14: **Accuracy and calibration performance on novel classes across 10 fine-grained classification benchmarks using RN-50 and ViT-B/32**. We report top-1 accuracy (Acc) and Expected Calibration Error (ECE) for CoOpZhou et al. (2022b), a prompt-tuning strategy, evaluated with different backbones.

| Method | | Calt | Pets | Cars | Flow | Food | Air | SUN | DTD | Euro | UCF | Avg |
|---|---|---|---|---|---|---|---|---|---|---|---|---|
| **CoOp-RN50** Zhou et al. (2022b) | | | | | | | | | | | | |
| CoOp-RN50 Zhou et al. (2022b) | Acc. | 87.27 | 92.11 | 57.84 | 61.87 | 82.55 | 18.72 | 64.46 | 41.67 | 34.15 | 55.07 | 11.38 |
| | ECE | 3.57 | 2.10 | 8.08 | 10.24 | 0.86 | 18.75 | 9.07 | 25.47 | 22.18 | 13.47 | 11.38 |
| ZS-Norm Murugesan et al. (2024) | Acc. | 88.39 | 89.97 | 57.86 | 59.93 | 81.52 | 20.34 | 65.77 | 35.79 | 40.08 | 57.37 | 59.70 |
| | ECE | 4.30 | 3.09 | 2.43 | 5.99 | 6.33 | 4.37 | 2.64 | 10.97 | 9.24 | 2.98 | 5.23 |
| Penalty Murugesan et al. (2024) | Acc. | 88.75 | 93.38 | 61.37 | 83.63 | 20.22 | 67.45 | 45.21 | 31.57 | 58.16 | 3.01 | 61.05 |
| | ECE | 3.05 | 7.94 | 1.7 | 4.81 | 8.24 | 10.02 | 2.92 | 9.96 | 11.67 | 3.01 | 6.33 |
| **Ours** | Acc. | 87.52 | 93.08 | 59.88 | 61.47 | 82.50 | 21.66 | 64.94 | 39.05 | 42.75 | 53.94 | 60.68 |
| | ECE | 2.82 | 2.68 | 3.43 | 4.13 | 1.72 | 5.36 | 4.53 | 8.83 | 10.93 | 10.16 | 5.46 |
| **CoOp–ViT-B/32** Zhou et al. (2022b) | | | | | | | | | | | | |
| CoOp–ViT-B/32 Zhou et al. (2022b) | Acc. | 92.25 | 94.00 | 60.04 | 60.31 | 85.16 | 22.12 | 68.98 | 47.95 | 56.47 | 63.57 | 64.63 |
| | ECE | 3.29 | 2.30 | 8.53 | 13.28 | 1.33 | 16.75 | 8.57 | 19.74 | 17.18 | 10.32 | 10.13 |
| ZS-Norm Murugesan et al. (2024) | Acc. | 91.56 | 93.46 | 59.01 | 54.09 | 84.50 | 21.64 | 70.14 | 43.44 | 52.04 | 65.98 | 63.59 |
| | ECE | 2.62 | 7.94 | 4.05 | 10.94 | 6.86 | 5.59 | 1.34 | 17.82 | 14.8 | 3.13 | 7.51 |
| Penalty Murugesan et al. (2024) | Acc. | 92.39 | 96.12 | 59.83 | 53.78 | 86.58 | 23.90 | 70.52 | 45.45 | 44.66 | 61.77 | 63.560 |
| | ECE | 4.41 | 6.93 | 2.66 | 7.31 | 5.23 | 8.63 | 2.19 | 8.77 | 10.71 | 4.22 | 6.11 |
| **Ours** | Acc. | 91.41 | 93.48 | 61.10 | 61.65 | 84.22 | 23.08 | 70.59 | 50.23 | 56.55 | 64.02 | 65.63 |
| | ECE | 2.23 | 2.15 | 4.33 | 5.92 | 0.74 | 9.90 | 3.72 | 14.30 | 9.14 | 5.80 | 5.82 |

Figure 6 shows that the Text Momentum-Matching loss better preserves the geometric structure of CLIP's pretrained embedding space by aligning the statistical moments of tuned and frozen text embeddings, compared to $\ell_1$ alignment or Orthogonality-based class-wise dispersion Sharifdeen et al. (2025) on novel classes.

## A.9 REPRODUCIBILITY STATEMENT

We make every effort to enable full reproduction of our results. The model and training procedrure are specified in Method section 3, including the two losses mean–variance margin regularization and text moment-matching with exact formulas and the full objective. Datasets, splits, and evaluation protocol (few-shot with base/novel classes) are described in (Experiments4: Datasets and Evaluation Metrics), and the exact hyperparameters, hardware, seeds, and implementation choices are consolidated in the Appendix A.4. We report averages over 3 random seeds and provide additional robustness, variance, and distribution-shift results in AppendixA.6, A.2. Prompt templates and initialization variants used in all runs appear in Supplementary A.5; backbone ablations are in A.7.

### A.10 The Use of Large Language Models (LLMs)

We made limited use of large language models to enhance the clarity and readability of the text. They were not involved in the conception of ideas, experiment design, analysis, or the production of results.

### A.11 Limitations

Despite these advances, our work has several limitations. First, our moment-matching approach relies on batch statistics, which may become unstable with very small batch sizes or highly imbalanced class distributions. Second, our method introduces additional hyperparameters that require tuning, potentially increasing computational overhead during development.

