# OpenReview forum: "TOWARDS CALIBRATING PROMPT TUNING OF VISION- LANGUAGE MODELS"
_ICLR.cc/2026/Conference — ICLR 2026 Conference Withdrawn Submission_

### Official Review · Reviewer_Yh9i · 2025-10-26

**Soundness:** 3
**Presentation:** 2
**Contribution:** 3
**Rating:** 4
**Confidence:** 5

**Summary:**

The paper addresses dual miscalibration in fine-tuned CLIP. The authors first identify the asymmetric boundary distortion in the fine-tuning phase, which modifies class decision boundaries and leads to miscalibration. To mitigate this issue, they propose a dual-regularization calibration framework consisting of mean–variance margin regularization and a moment-matching. Extensive experiments on multiple datasets show consistent calibration improvement on both base and novel classes.

**Strengths:**

1.	The problem formulation is clear. The calibration degradation occurs differently for base and novel classes in fine-tuning, and they propose a dual calibration framework that addresses both simultaneously.

2.	The method directly aligns the motivation and miscalibration types. The two complementary regularization terms reflect the authors’ analysis of different miscalibration types: a mean-variance margin loss to stabilize logit distributions and a text moment-matching loss to preserve the semantic geometry crucial for novel class generalization.

3.	The experimental results are comprehensive and generally support the author’s claims. The proposed method consistently improves ECE across both base and novel classes without sacrificing accuracy. Moreover, the ablation in Table 3 clearly shows that both regularizations are effective.

**Weaknesses:**

1．	The description of the motivation lacks clarity and support. The paper introduces the concept of "asymmetric boundary distortions" and claims that the underconfidence of the base class reflects "reduced margins". This claim is not clearly supported by the visualizations (e.g., Figure 5(c) and Figure 6.

2．	The method design is heuristic. The author proposes matching the first and second-order moments for the text moment-matching loss. However, they do not provide a justification for why this choice is optimal compared to other potential distributional alignment methods.

3．	The paper lacks discussion and comparison with several recent training-based calibration methods [1-2], which also aim to improve calibration during the fine-tuning phase.

4．	The claim that the method "preserves critical semantic relationships" is not fully supported. It is hard to convincingly conclude such a claim from the t-SNE visualization (Figure 6).

5．	As acknowledged in the limitations, this dependency may harm stability in low-shot settings. However, it does not evaluate performance with very small batch sizes or training shots.

[1] Wang S, Li Y, Wei H. Understanding and mitigating miscalibration in prompt tuning for vision-language models. ICML, 2025.

[2] Oh C, Lim H, Kim M, et al. Towards calibrated robust fine-tuning of vision-language models. NeurlPS 2024, 37: 12677-12707.

**Questions:**

Does applying only one of the loss components introduce a calibration trade-off between base and novel classes? It is still unclear how $L_{Margin}$  or $L_{mom}$  individually affect the calibration for both classes.

---

### Official Review · Reviewer_QCvY · 2025-10-27

**Soundness:** 3
**Presentation:** 3
**Contribution:** 2
**Rating:** 4
**Confidence:** 2

**Summary:**

This paper proposes a novel and effective training-time framework that jointly addresses the dual confidence calibration problem in prompt-tuned VLMs through boundary stabilization and moment matching. The method is model-agnostic and plug-and-play, enabling more reliable and trustworthy predictions across a variety of settings without modifying the underlying model architecture.

**Strengths:**

1. The mean-variance boundary regularizer, applied in the logit space, and the text moment matching loss, applied in the embedding space, are conceptually complementary.

2. The method is plug-and-play and highly adaptable.

3. It is evaluated on 11 datasets and 7 mainstream prompt-tuning methods, providing strong evidence of its effectiveness.

**Weaknesses:**

1. The method introduces several hyperparameters. Although the paper claims in Appendix A.4 that a fixed set of default values works well across all experiments, this may obscure potential sensitivity to hyperparameters on specific datasets or tuning methods.

2. The authors state that the method is a training-time regularization that does not increase inference cost. However, it does incur additional computational and memory overhead during training, as each training step requires computing the original CLIP text embeddings for moment matching and performing extra boundary calculations. A quantitative analysis of training time and memory usage would be valuable.

3. In more extreme low-resource scenarios, such as 1-shot or 2-shot settings, it is unclear whether the batch-statistics-based moment matching loss remains stable and effective when training data is extremely scarce.

4. Why does prompt tuning lead to such asymmetric boundary changes, with seen-class boundaries tightening and unseen-class boundaries expanding? Is this due to overfitting, or does prompt learning introduce some inherent bias?

**Questions:**

See the Weakness.

---

### Official Review · Reviewer_y85K · 2025-10-30

**Soundness:** 3
**Presentation:** 3
**Contribution:** 3
**Rating:** 6
**Confidence:** 4

**Summary:**

This paper proposes a plug-and-play prompt-tuning calibration framework that requires no architectural changes. It targets the dual-calibration problem in large vision-language models such as CLIP: under-confidence on base classes and over-confidence on novel classes after prompt tuning. The core idea is to introduce two complementary regularization during training: (1) a mean–variance margin penalty and (2) a text moment-matching matching loss. Experiments on 11 datasets and 7 prompt-tuning backbones show significant ECE reductions while maintaining or improving accuracy.

**Strengths:**

1. Originality: First work to explicitly formulate the “base-class under-confidence + novel-class over-confidence” dilemma as a dual-calibration problem; combines statistical margin constraints with moment matching, differing from previous temperature-scaling or post-hoc normalization methods.
2. Quality: Comprehensive experiments covering multiple datasets, prompt-tuning methods, and calibration metrics; consistently outperforms existing calibration baselines.
3. Clarity: Clear problem statement, motivation, equations, and algorithm diagrams; ablation studies quantify the contribution of each regularization; appendix provides hyper-parameters, templates, and reproducibility details.
4. Significance: Prompt tuning is now the default paradigm for adapting VLMs, but unreliable confidence impedes deployment in safety-critical domains. The proposed method is inference-free and plug-and-play, making it immediately usable in healthcare, autonomous driving, etc.

**Weaknesses:**

1. When the batch size is small or class distribution is extremely imbalanced, the moment-matching loss may become unstable; the paper does not offer a concrete remedy.
2. The evaluation only reports global calibration metrics such as ECE; it does not analyze the confidence distribution of mis-predictions, e.g., whether certain classes tend to be over-confident, or whether semantically similar class pairs are more prone to miscalibration.
3. Although the title claims to address prompt-tuning calibration for vision-language models, all experiments are conducted only on CLIP; no evidence is provided that the approach generalizes to other VLMs.

**Questions:**

See the Weaknesses.

---

### Official Review · Reviewer_pKd7 · 2025-10-31

**Soundness:** 3
**Presentation:** 3
**Contribution:** 2
**Rating:** 4
**Confidence:** 4

**Summary:**

This paper addresses the poor confidence calibration in prompt-tuned vision-language model CLIP. The authors propose a calibration framework that preserves embedding geometry through two regularizers: a mean–variance margin penalty and a text moment-matching loss. Extensive experiments across multiple datasets show improved predictive reliability and reduced calibration error without compromising generalization.

**Strengths:**

1. Addressing the issue of poor confidence calibration in prompt-tuned vision-language models is both timely and meaningful. This paper presents a promising approach by introducing two well-designed regularizers that enhance calibration while preserving embedding geometry. The extensive experiments on the CLIP model demonstrate the method’s effectiveness in improving the calibration performance.

2. The paper is well structured and clearly written, making it easy to follow.

**Weaknesses:**

1. In the introduction, providing more concrete illustrations or examples of the potential issues caused by confidence miscalibration would help clarify and better highlight the significance and contribution of this work.

2. The scope and applicability of the proposed method appear largely limited to the CLIP model. Evaluating the scalability of the proposed regularizers on other vision-language frameworks (e.g., BLIP2 [1]) and larger-scale models (e.g., Qwen3-VL-4B/8B [2]) would strengthen the relevance and potential for real-world applications. Limiting the study to CLIP constrains the broader impact of the approach.

3. As shown in Figure 2, for base-class evaluation, the improvement from adding $L_{moon}$ is not clearly demonstrated compared with $L_{CE}+L_{margin}$. Moreover, for novel classes, the proposed regularizers appear less effective in mitigating overconfidence (where predicted confidence exceeds accuracy) compared with the baseline cross-entropy loss. Additional analysis or discussion is needed to clarify these observations.

[1] Li, J., Li, D., Savarese, S., & Hoi, S. (2023, July). Blip-2: Bootstrapping language-image pre-training with frozen image encoders and large language models. In International conference on machine learning (pp. 19730-19742). PMLR.

[2] https://github.com/QwenLM/Qwen3-VL?tab=readme-ov-file

**Questions:**

1. As shown in Figure 2, for base-class evaluation, the improvement from adding $L_{moon}$ is not clearly demonstrated compared with $L_{CE}+L_{margin}$. Moreover, for novel classes, the proposed regularizers appear less effective in mitigating overconfidence (where predicted confidence exceeds accuracy) compared with the baseline cross-entropy loss.

---

### Note · Authors · 2025-11-13

I have read and agree with the venue's withdrawal policy on behalf of myself and my co-authors.